



# Quantitation of nine alkyl amines in atmospheric samples:
# Separating structural isomers by ion chromatography
Bryan K. Place[1, 2], Aleya T. Quilty[2], Robert A. Di Lorenzo[1], Susan E. Ziegler[2] and Trevor C.
VandenBoer[2]
[1]Department of Chemistry, Memorial University, St. John's, NL, Canada
[2]Department of Earth Sciences, Memorial University, St. John's, NL, Canada
Correspondence to: T. C. VandenBoer (tvandenboer@mun.ca)
**Abstract:** Amines are important drivers in particle formation and growth, which has implications for Earth's
climate. In this work, we developed an ion chromatographic (IC) method using sample cation-exchange
preconcentration for separating and quantifying the nine most abundant atmospheric alkyl amines
(monomethylamine ($MMAH^+$), dimethylamine ($DMAH^+$), trimethylamine ($TMAH^+$), monoethylamine
($MEAH^+$), diethylamine ($DEAH^+$), triethylamine ($TEAH^+$), monopropylamine ($MPAH^+$), isomonopropylamine
($iMPAH^+$), and monobutylamine ($MBAH^+$)). Further, the developed method separates the suite of amines from
five common atmospheric inorganic cations ($Na^+$, $NH_4^+$, $K^+$, $Mg^{2+}$, $Ca^{2+}$). All 14 cations are greater than 95%
baseline resolved and elute in a runtime of 35 minutes. This paper describes the first successful separation of
$DEAH^+$ and $TMAH^+$ by IC and achieves separation between three structural isomer pairs, providing specificity
not possible by mass spectrometry. The method detection limits for the alkyl amines are in the picogram per
injection range and the method precision ($\pm1$ σ) analyzed over 3 months was within 16 % for all the cations. The
performance of the IC method for atmospheric application was tested with biomass-burning (BB) particle
extracts collected from two forest fire plumes in Canada with large loadings of $K^+$. In extracts of size-resolved
BB samples from an aged plume we detected and quantified $MMAH^+$, $DMAH^+$, $TMAH^+$, $MEAH^+$, $DEAH^+$ and
$TEAH^+$ in the presence of $Na^+$, $NH_4^+$, and $K^+$ at amines to inorganic cations molar ratios ranging from 1:2 to
1:1000. Quantities of $DEAH^+$ and $DMAH^+$ of 0.2 – 200 ng m$^{-3}$ and 3 – 1200 ng m$^{-3}$, respectively, were present
in the extracts and an unprecedented amines to ammonium molar ratio greater than one was observed in particles
with diameters spanning 56 – 180 nm. Extracts of respirable fine mode particles ($PM_{2.5}$) from a summer forest
fire in British Columbia in 2015 were found to contain $iMPAH^+$, $TMAH^+$, $DEAH^+$ and $TEAH^+$ at molar ratios of
1:300 with the dominant cations. The amines to ammonium ratio in a time-series of samples never exceeded 0.15
during the sampling of the plume. These results demonstrate the robustness and sensitivity of the developed
method when applied to the complex matrix of biomass burning particle samples. The detection of multiple alkyl
amines in the analyzed BB samples indicates that this speciation and quantitation approach can be used to
constrain BB emission inventories and the biogeochemical cycling of these reduced nitrogen species.



## 1 Introduction

Particles in the atmosphere can modulate climate through their direct and indirect effect on the radiative balance of Earth's atmosphere (Boucher et al., 2013; Lohmann and Feichter, 2005). This potential warming or cooling effect of particles represents the greatest uncertainty in Earth's radiative forcing (Myhre et al., 2013). Additionally, particles with a diameter of 2.5 μm or less ($PM_{2.5}$) have been classified as carcinogens (IARC), and are estimated to be responsible for 3 million deaths annually worldwide (Stephen et al., 2012). Thus understanding the quantities, and the chemical and physical nature of the species involved in the formation and growth of new particles is of paramount importance.

Recent work has shown that organic compounds may contribute considerably to particle nucleation (Ehn et al., 2014; Ortega et al., 2016; Tröstl et al., 2016; Willis et al., 2016). In particular, the need to measure and quantify gaseous atmospheric alkyl amines has gained interest because of their exceptional ability to partake in atmospheric particle formation. Multiple laboratory investigations have shown the nucleation potential of methyl- and ethyl-substituted amines through gaseous acid-base chemistry reactions (Almeida et al., 2013; Angelino et al., 2001; Berndt et al., 2010; Berndt et al., 2014; Bzdek et al., 2010; Bzdek et al., 2011; Erupe et al., 2011; Jen et al., 2016; Lloyd et al., 2009; Murphy et al., 2007; Qiu et al., 2011; Silva et al., 2008; Smith et al., 2010; Wang et al., 2010a; Wang et al., 2010b; Yu et al., 2012; Zhao et al., 2011; Zollner et al., 2012). Theoretical calculations and studies have also found that amines have a high disposition to form atmospheric nanoparticles (Barsanti et al., 2009; Kurtén et al., 2008; Loukonen et al., 2014; Loukonen et al., 2010; Nadykto et al., 2015; Ortega et al., 2012). From these works, alkyl amines have been shown to form clusters via neutralization reactions at rates up to three orders of magnitude greater than ammonia (Almeida et al., 2013; Berndt et al., 2010; Bzdek et al., 2011; Kurtén et al., 2008; Loukonen et al., 2010; Nadykto et al., 2015), and readily exchange with ammonia in already formed ammonium-bisulfate molecular clusters (Bzdek et al., 2010; Lloyd et al., 2009; Qiu et al., 2011). These studies suggest that alkyl amines can compete with ammonia to form particles even though they have been quantified at mixing ratios that are three or more orders of magnitude less in the atmosphere (Chang et al., 2003; Ge et al., 2011; Schade and Crutzen, 1995). Atmospheric measurements made during new particle formation events have further confirmed that alkyl amines participate in particle formation at ambient concentrations and that these species may be present in most atmospheric particles (Creamean et al., 2011; Dall'Osto et al., 2012; Hodshire et al., 2016; Kulmala et al., 2013; Kürten et al., 2016; Ruiz-Jimenez et al., 2012; Smith et al., 2010; Tao et al., 2016).

Alkyl amine emissions to the atmosphere arise from both natural and anthropogenic sources (Ge et al., 2011). Short-chain alkyl amines such as the methylated and ethylated amines are predominantly reported in emission inventories and measurements show that atmospheric alkyl amines are prevalent in ambient air across the globe, especially in the particle phase (Ge et al., 2011). For example, methyl and ethyl amines were measured by an aerosol time-of-flight mass spectrometer (ATOFMS) at both rural and urban sites all across Europe (Healy et al., 2015). In particular, these amines have been measured in substantial quantities near animal husbandry operations (Kuhn et al., 2011; Lunn and Van de Vyver, 1977; Rabaud et al., 2003; Schade and Crutzen, 1995; Sorooshian et al., 2008), fisheries (Seo et al., 2011), and sewage-waste treatment facilities (Leach et al., 1999). Other anthropogenic sources include tobacco smoke (Schmeltz and Hoffmann, 1977), automobiles (Cadle and Mulawa, 1980) and cooking (Rogge et al., 1991; Schauer et al., 1999). The ocean is estimated to be the largest natural



source of alkyl amines, where they are released as degradation products (Ge et al., 2011; Gibb et al., 1999a; Gibb
et al., 1999b). Aliphatic amines have also been detected in smoldering stage biomass burning plumes and are
estimated to represent a quarter of global methylated amine emissions (Lobert et al., 1990; Schade and Crutzen,
4   1995).

Real-time speciation and quantitation of atmospheric amines in the particle and gas phase can be difficult
because alkyl amines are commonly found at or below parts per trillion by volume (pptv) mixing ratios in the
atmosphere (Ge et al., 2011). Furthermore, the atmospheric matrix can be complex and ubiquitous atmospheric
species can cause matrix effects for various analytical methods. Being able to chromatographically resolve
ammonium from the alkyl amines represents a major challenge when sampling the gas phase, since quantities of
ammonia are often orders of magnitude greater than the alkyl amines (Chang et al., 2003; Ge et al., 2011; Schade
and Crutzen, 1995). Quantifying amines in particle samples, however, presents a greater challenge due to
possible ionic interferences from sodium, potassium, ammonium, magnesium and calcium concentrations
dependent on the particle source characteristics and the measurement location (Ault et al., 2013; Kovac et al.,
2013; Sobanska et al., 2012; Sun et al., 2006). Particles also frequently contain complex high molecular weight
organic compounds, which can cause further matrix effects during separation or direct analysis (Di Lorenzo and
Young, 2016; Saleh et al., 2014).
Achieving full speciation of alkyl amines is important because the nucleation potential of amines has been
shown to increase with basicity (Berndt et al., 2014; Kurtén et al., 2008; Yu et al., 2012). For example, although
monopropylamine (MPA) and trimethylamine (TMA) are structural isomers of one another, MPA is likely to be
a more potent nucleator due to its stronger basicity.  The suite of alkyl amines that have been commonly detected
in the atmosphere contains multiple structural isomers (e.g. monoethylamine (MEA) and dimethylamine
(DMA)), making it difficult to speciate the amines using mass spectrometry (MS) without prior separation.
Multiple field investigations sampling atmospheric particles using MS analysis have reported the detection of
amine ion peaks but have been unable to assign them to a specific amine (Aiken et al., 2009; Denkenberger et
al., 2007; Silva et al., 2008; Yao et al., 2016).  Derivatization of alkyl amines coupled with HPLC or GC
separation has been reported to aid in separation and quantitation of amine species (Akyüz, 2007; Huang et al.,
2009; Fournier et al., 2008; Key et al., 2011; Possanzini and Di Palo, 1990). However, these approaches are time
consuming, require optimization of reaction conditions, and employ phase separations, which use large
quantities of consumables, reagents, and solvents.  Capillary electrophoresis has also been employed for aqueous
amine separation, however in either case derivatization was required (Dabek-Zlotorzynska and Maruszak, 1998)
or the separation of atmospherically relevant cations was not addressed (Fekete et al., 2006). The use of ion
chromatography to directly separate and quantify atmospheric alkyl amines has been demonstrated (Chang et al.,
2003; Dawson et al., 2014; Erupe et al., 2010; Huang et al., 2014; Li et al., 2009; Murphy et al., 2007;
VandenBoer et al., 2012; Verriele et al., 2012), yet the established IC methods struggle with coeluting cations
(Huang et al., 2014; Murphy et al., 2007; VandenBoer et al., Verriele et al., 2012) or they do not address a full
suite of atmospherically relevant alkyl amines and inorganic cations (Chang et al., 2003; Dawson et al., 2014;
Erupe et al., 2010; Li et al., 2009).
In this work we demonstrate the separation and quantitation of the nine most abundant atmospheric alkyl amines
and six inorganic cations through the use of ion chromatography. We show i) the separation method approach to





maximizing peak resolution in the context of real-time atmospheric sampling and analysis; ii) the effects of column temperature on amine coelution; iii) the method precisions, accuracies, sensitivities and limits of detection (LODs) for all alkyl amine and inorganic cations; and iv) application of the method to the complex matrix of atmospheric biomass burning particle extracts to demonstrate method sensitivity and robustness.

## 2    Methods

### 2.1    Chemicals and materials

Inorganic cation stock solutions were prepared from a primary mixed cation standard concentrate (Dionex six-cation II, Lot #150326, Thermo Scientific, Waltham, MA, USA) consisting of $Li^+$, $Na^+$, $NH_4^+$, $K^+$, $Mg^{2+}$ and $Ca^{2+}$ chloride salts. Alkyl amines (MMA (40% w/w), DMA (40% w/w), TMA (25 % w/w), MEA (70 % w/w), DEA (>99% w/w), TEA (>99% w/w), MPA ($\geq$ 99% w/w), iMPA ($\geq$ 99% w/w) and MBA (99.5% w/w)) were purchased from Sigma-Aldrich (Oakville, ON, Canada). Calibration standards were prepared by serial dilution in 18.2 M$\Omega$ ultrapure deionized water (Barnstead Nanopure Infinity, Thermo Scientific, Waltham, MA, USA). Since these alkyl amine species will largely be protonated in solution we will denote each of these species in their cationic form (i.e. $NR_3H^+$) henceforth when referring to the condensed phase.

### 2.2    Ion Chromatography

A ThermoScientific ICS-2100 Ion Chromatography System (Thermo Scientific, Mississauga, ON, Canada) utilizing Reagent-Free Ion Chromatography (RFIC$^{TM}$) components was used to develop the separation of the selected amines and inorganic cations. A ThermoScientific methanesulfonic acid (MSA) eluent generator cartridge (EGC) III (P/N: 074535) was used in conjunction with an ultrapure deionized water reservoir to supply the eluent mobile phase with $H_3O^+$ ions as the competing exchanger. A continuously regenerated trap column (CR-CTC II, P/N: 066262) was attached to the eluent cartridge in series to remove cation contaminants from the eluent prior to analysis, thereby reducing method detection limits. Samples were preconcentrated on a cation exchange column (TCC-ULP1; 5 x 23 mm, P/N: 063783) using a ThermoScientific AS-DV autosampler to deliver the desired volume. Concentrated analytes were separated using ThermoScientific CG19 (4 x 50 mm, P/N: 076027) and CS19 (4 x 250 mm, P/N: 076026) guard and analytical cation-exchange columns. The column effluent was passed through a suppressor operating in recycle mode (CSRS 300, 4 mm) prior to detection of the analytes using a DS6 heated conductivity cell thermostated at 30 °C. The eluent conductance was recorded at 5 Hz and the chromatographic peaks were analyzed using the Chromeleon 7 software package. A ThermoScientific CG15 (4 x 50 mm, P/N: 052200) guard column was added inline later to attempt further improvement in analyte separation.

### 2.3    CS19 separation optimization

The gradient program used for the separation of methylamines, ethylamines, other alkyl amines and six inorganic cations on the CS19 cation-exchange column was optimized by combining analyte separation parameters from multiple isocratic elution runs at varying MSA concentrations (1 - 16 mM) and mobile phase flow rates (0.75 -





1.25 ml min$^{-1}$) at a column temperature of 30 °C. Maximum peak resolution was optimized using an eluent gradient program and  the column temperature was increased to resolve co-eluting peaks (see Sect. 3.1.1).

Optimal separation of a suite of 15 cations was achieved using a mobile phase flow rate of 1.25 ml min$^{-1}$ and a column temperature of 55 °C. The eluent gradient program is as follows: an initial MSA concentration of 1 mM held for 20 minutes, a step increase to 4 mM followed immediately by an exponential ramp to 10 mM over 10 minutes (Chromeleon curve factor = 7). The final concentration of 10 mM was held for an additional 5 minutes, yielding a total run time of 35 minutes. The IC was returned to initial conditions and re-equilibrated as the next sample was prepared for injection by the AS-DV.

## 2.4    Quality Assurance and Quality Control

Standards were prepared using Class A Corning polymethylpentene 50 ($\pm$ 0.06) ml volumetric flasks that were rinsed 4 times with ethanol and 8 times with ultrapure water prior to use. Standards were stored in 60 ml brown Nalgene polypropylene bottles that were pre-cleaned in a 10 % HCl bath, followed by 8 sequential rinses with distilled and ultrapure water, respectively. The mixed amine standards and mixed inorganic cation standards were prepared separately and each cation standard set was composed of five calibration standards and two check standards. Ranges and related parameters are denoted by mass injected, as the preconcentration column negates the effect of volume. All amine calibration standards had a mass calibration range of 5-500 ng.. The mass range for each inorganic cation calibration is as follows: Li$^+$ (0.82-160 ng), Na$^+$ (7.8-160 ng), NH$_4^+$ (8.4-170 ng), K$^+$ (26-520 ng), Mg$^{2+}$ (6.4-130 ng), and Ca$^{2+}$ (18-360 ng).

Method precision for each methyl and ethyl amine cation was determined using standard calibration curves (n = 9) injected across five different days spanning three months. The precision for the propyl and butyl amines was determined using two standard calibration curves analyzed over one month. Method precision for Li$^+$, Na$^+$, NH$_4^+$ and K$^+$ was assessed using calibrations (n = 6) from three separate days spanning two months. Precision for each cation was calculated using the standard deviation (σ) in the slope of the linear calibration curves. Check standards positioned between the two highest and the two lowest calibration standards for each cation were used to determine method accuracy across the calibration range. The low check standard was 15 times greater than the lowest standard and the high check standard was 150 times higher than the lowest standard. Accuracy was determined by the percent relative error between the known and calculated concentrations of the check standards. The limits of detection (LOD) for the singly-charged inorganic cations (n = 4) and methyl and ethyl amines (n = 5) were determined using calibration standard and calibration blank chromatograms from three or more separate days. The LODs for the propyl and butyl amines were determined using calibration standard and blank chromatograms from two separate days. The LODs are reported as concentrations resulting in a signal peak height to background noise ratio of three. The background noise was determined using the standard deviation of the conductance signal that fell within the retention time window for each analyte in their respective calibration blank chromatograms. Discussion of the analytical performance of the CS19 gradient program is presented in Sect. 3.1.2. and divalent cations in Sect. 3.1.5.





## 2.5 Size-resolved BB sample analysis

A size-resolved particle sample from a BB plume was collected using a NanoMOUDI II (Nano micro-orifice uniform-deposit impactor, model 122-R, MSP Corp., Shoreview, MN, USA) in St. John's, Newfoundland on July 6, 2013. Satellite images of the plume smoke, HYSPLIT back trajectories, as well as, measured $PM_{2.5}$ concentrations reported by Environment and Climate Change Canada (Di Lorenzo and Young, 2016) indicate that these plumes originated from boreal forest fires in northern Quebec and Labrador on July 4, 2013 and travelled via Labrador and the Gulf of St. Lawrence to the sampling site. The nanoMOUDI samples were collected on 13 aluminum substrate stages in to size-resolved bins of atmospheric particles with a diameter range spanning 0.010 –18 μm. Air was sampled continuously for 25.5 hours at a flow rate of 30 L min$^{-1}$. A sub-sample of each aluminum substrate (10 % of the total substrate area) was extracted into a glass vial with 5 mL ultrapure deionized water by sonication (VWR Scientific Products/Aquasonic 150HT, Ultrasonic Water Bath) for 40 minutes. The extracts were filtered using a 0.2 μm polytetrafluoroethylene (PTFE) filter and stored in polypropylene vials at 4 °C prior to analysis by IC within 24 hours.

## 2.6 BC fire sample analysis

The full method for the collection and extraction of BB particle samples collected during July wildfires in British Columbia is detailed in Di Lorenzo et al. (2016). Briefly, $PM_{2.5}$ samples were collected at two sites approximately 100 kilometers east of the biomass-burning location. The first site was located in Burnaby/Kensington Park (BKP) and the second was in North Vancouver/Second Narrows (NVSN). The particle samples were collected using real-time beta attenuation particle monitors (5030 SHARP Monitor at the BKP site, 5030i SHARP monitor at the NVSN site, Thermo Fisher Scientific, Waltham, MA, USA) at a 16.67 L min$^{-1}$ flow rate in 8-hour intervals. Particles were collected on glass microfiber filter tape and stored at -20 °C until extracted. Approximately 37% of each filter spot area was placed into a polypropylene vial with 10 ml of deionized water, and sonicated for 40 minutes. The extracts were filtered with PTFE syringe filters (3 mm diameter, 0.2 μm pore size, VWR International, Radnor, Pennsylvania, USA) and diluted by a factor of five with ultrapure deionized water before being injected on the IC.

## 3 Results and Discussion

### 3.1 Analytical method performance of CS19 cation exchange column

#### 3.1.1 Separation approach and optimization of parameters

Our approach to separation involved injecting the highest mixed inorganic cation and mixed amine standards for the expected working range (0.1 – 2.5 μg ml$^{-1}$) at static flow rates (0.75 ml min$^{-1}$, 1 ml min$^{-1}$, and 1.25 ml min$^{-1}$) while systematically increasing the isocratic eluent concentration (4 mM - 16 mM). The quality of each isocratic method was assessed by calculating the peak-to-peak resolution ($R_s$) using the retention time ($t_R$) and peak width at base (w) determined from the highest standard for each pair of cations following Eq. (1):





$R_S = \frac{2(t_{R2} - t_{R1})}{w_2 + w_1}$,                                                  (1)
Using the upper limit of the expected working range for all analytes therefore provides a lower limit on peak-to-
peak resolution between these species. The peak-to-peak resolutions of the isocratic methods run using a 0.75 ml
min⁻¹ and 1.25 ml min⁻¹ flow rate for the selected inorganic and alkyl amine cations are presented in Fig. S1 and
S2. Peak-to-peak resolution between all peaks increased as the mobile phase ionic strength was lowered when
the flow rate was held constant. This is in agreement with Eq. (2), the fundamental resolution equation, which
describes peak-to-peak resolution in terms of an efficiency factor (N), retention factor (k), and a selectivity factor
($\alpha$).
$R = \left(\frac{\sqrt{N}}{4}\right)\left(\frac{k}{k+1}\right)\left(\frac{\alpha-1}{\alpha}\right)$,                                        (2)
With low mobile phase ionic strength, the retention factor of the analytes is expected to increase, leading to
greater resolution, consistent with our observations. In contrast, the effect of flow rate on peak resolution is non-
intuitive and must be obtained empirically. Lower flow rates increase the retention factor, which in turn
increases resolution. However, an increase in mobile phase flow rate has a competing effect on the efficiency
factor in Eq. (2). The efficiency term is governed by the theoretical plate height (H) as described by the Van
Deemter equation (Eq. (3)), which highlights the competing effect of flow rate ($\mu$) on peak resolution:
$H = A + \frac{B}{\mu} + C\mu$,                                                    (3)
Figures S1 and S2 show no loss in peak resolution when using a higher flow rate (1.25 ml min⁻¹ vs 0.75 ml min⁻
¹). To confirm that there was no loss in efficiency at higher flows, Van Deemter plots were created, using
MMAH⁺ and TEAH⁺ as representative early and late-eluting species, by plotting theoretical plate height versus
flow rate (Fig. S3). To do this, the theoretical plate heights described in Eq. (3) were calculated using Eq. (4),
which relates H to column length (L), $t_R$ and w. The A, B and C terms of Eq. (3) were then determined by
solving a system of equations using the calculated H values for MMAH⁺ and TEAH⁺ at three isocratic flow
rates.
$\frac{L}{H} = 16\left(\frac{t_R}{w}\right)^2$,                                                (4)
The Van-Deemter plots for MMAH⁺ and TEAH⁺ show no sacrifice in resolution when operating at higher flow
rates and low eluent concentrations. The resolution between Mg²⁺ and Ca²⁺ as well as between TMAH⁺ and
TEAH⁺ improved at the higher flow rate for all isocratic eluent concentrations. This was due to a decrease in
peak width from diffusion band broadening. Furthermore, there was little to no sacrifice in resolution for all
other cation peak pairs when operating at a higher flow rate. Of particular note, utilizing a 4 mM MSA isocratic
separation at a 1.25 ml min⁻¹ flow rate instead of a 0.75 ml min⁻¹ flow resulted in a runtime that was 20 minutes
shorter, which improves the applicability of IC for near-real-time analysis of hourly to bi-hourly atmospheric
sample collection timescales. A shorter run time also improves the method throughput capacity for offline
analyses and reduces total eluent consumption. For these reasons, the faster flow rate was selected in designing
and optimizing a gradient program. Further isocratic methods using lower MSA concentrations (1 mM and 2
mM) were run at a 1.25 ml min⁻¹ flow rate to quantify values of peak-to-peak resolution for the inorganic cations



and alkyl amines before approaching a gradient method (Fig. S1 and S2). An increase in resolution greater than
one was observed for all analyte pairs aside from DEAH$^+$/TMAH$^+$ when using a 1 mM MSA eluent
concentration.
All gradient methods that were tested started with a 1 mM hold, followed by a step-wise increase and/or ramp to
higher eluent concentrations at a column temperature of 30 °C. By combining the best isocratic separations for
various pairs of cation analytes sequentially, iterative modifications were used to improve resolution based on
Eq. (1-3). The best separation method was selected from amongst the iterations and the column temperature was
then systematically increased to investigate if further improvement in peak-to-peak resolution was possible.
Temperature effects on separation efficiency in ion chromatography are thermodynamically complex (Hatsis and
Lucy, 2001; Kulis, K., 2004; Rey and Pohl, 1996), but typically result in increased peak resolution because of
improvements in mobile phase diffusivity, which increases the efficiency from Eq. (2). It has been demonstrated
that using higher temperatures can replicate the separation effects observed when adding an organic mobile
phase modifier (Hatsis and Lucy, 2001; Rey and Pohl, 1996). Figures 1a and 1b show gradient separations at 30
°C and 55 °C respectively. At 30 °C, K$^+$ and DMAH$^+$ overlap considerably (R$_S$ = 0.45) and DEAH$^+$ and TMAH$^+$
coelute. By increasing the column temperature to 55 °C, the extent of peak overlap between K$^+$ and neighboring
alkyl amine cations is noticeably reduced (R$_S$ > 1) and DEAH$^+$ and TMAH$^+$ are increasingly well resolved (R$_S$ =
1.48). The effect of temperature on the separation of the alkyl amines is demonstrated in Fig. 2, where the
separation of DEAH$^+$ with TMAH$^+$ is achieved above 50 °C. The temperature increase also results in lower
resolution between DMAH$^+$ and MEAH$^+$ from R$_S$ = 1.57 to R$_S$ = 1.08. A column temperature of 55 °C produced
peak-to-peak resolutions greater than a value of one between all alkyl amine cations in the final gradient method,
giving a 95 % separation between our target analytes and expected atmospheric interferences in the condensed
phase. The peak-to-peak resolutions are summarized in Table 1. These represent a lower-limit in peak resolution
since they were calculated using peak parameters at the upper limit of the working range, which was determined
based on typical mixing ratios or mass loadings expected for the analysis of atmospheric samples containing
these analytes.
The separation method produced in this work is able to overcome previously reported IC coelution difficulties
between DEAH$^+$ and TMAH$^+$ and between MEAH$^+$ and DMAH$^+$ (VandenBoer et al., 2012; Verriele et al.,
2011). Both DMA and TMA have been identified as dominant amines in emission studies, so it is important to
achieve accurate and specific quantitation of both species in gas and particulate atmospheric samples (Facchini et
al., 2008; Kuwata et al., 1983; Müller et al., 2009; Van Neste and Duce, 1987). Multiple field campaigns have
detected large quantities of MEA and DEA in ambient air as well (Facchini et al., 2008; Müller et al., 2009;
Sorooshian et al., 2009; Yang et al., 2005; Yang et al., 2004). In some cases clean up steps have been used to
alleviate IC interferences from common atmospheric cation species in the quantitation of amines despite the fact
that an 85% evaporation loss of the amines, in addition to the extra sample handling, was reported when using a
solid phase extraction clean up (Huang et al., 2014). The CS19 IC method reported here is able to separate the
most common atmospheric inorganic cations in addition to the six most common atmospheric amines. It can be
easily applied to water-soluble atmospheric gas and particulate samples since they can be directly analyzed -
without coelution or a clean-up step - with separation times of similar duration to many previously reported





methods, including those employing an online IC method (Huang et al., 2014; Murphy et al., 2007; VandenBoer
et al., 2012; Verriele et al., 2011).
**3.1.2**     **Method performance and comparison**
The performance statistics of the CS19 gradient method for each cation are summarized in Table 1. The method
shows high reproducibility, with method precisions better than 10 % for most analytes. The larger variability in
the $TMAH^+$ and $TEAH^+$ calibration curves ($\pm16$ % and $\pm11$ % respectively) over time could likely be driven by
their lower Henry's Law constants ($K_H$) in water (Christie and Crisp, 1967), resulting in volatilization losses
from standards. Concurrently, this variability could be driven by partitioning losses along the flow path,
particularly when the tri-subsituted amines reach the suppressor, which is not temperature-controlled. In future
investigations it may be worthwhile to acidify the standards to ensure the amines are maintained in their charged
form in the aqueous phase. Alternatively, to combat losses to neutral forms, use of a Salt Converter suppressor
accessory (ThermoScientific, SC-CSRS 300, P/N: 067530), which keeps weak electrolytes in a separated sample
fully protonated prior to their conductance measurement, may also aid in increasing long-term precision.
The limits of detection (LODs) for each analyte are reported in Table 1 as both a range and as the average LOD
($\pm$ 1σ). The LODs are reported in this manner to reflect the high day-to-day variability in the calculated LODs.
This variability may be driven by i) the purity of the deionized water used for eluent generation; ii) instrumental
baseline noise and trace contamination on the day of analysis; and iii) quality of labware cleaning prior to
preparation of calibration blanks. Outliers in the LOD dataset were found to result from trace contamination of
analytical labware, sampling vials, or from systematic errors made in the preparation of standards or injection of
samples on the IC (e.g. leaking autosampler caps, failing retention of concentrator column). The Grubb's test
was performed using a 95% confidence interval to statistically identify outliers from LOD data sets. Calculated
detection limits were determined to lie in the picogram per injection range for all analytes. The LODs for the
inorganic cations were 10 to 100 times lower than those of the alkyl amines. Our method shows high accuracy in
the upper range of the calibrations for the analytes, with accuracies ranging from 94 – 103 %. The accuracy was
much lower for each alkyl amine cation at the low end of the calibration range at concentrations approximately
1.5 times the LOQ. Quantitation near the method LOQ was more sensitive to small integration changes, which
affected the calculated peak area, even when performing integrations manually, and this resulted in greater
method error. This is a drawback inherent to IC since wide analyte peaks are a result of persistently large
stationary phase particle sizes, causing band broadening via longer flow paths and increased diffusion during
separation (i.e. the A and B terms in Eq. (3)). The low alkyl amine accuracies may also be driven by their air-
water partitioning properties, which could result in losses during sample handling and during sample injection.
The low-range accuracies for all inorganic cations, with the exception of ammonium, were still high (80 – 94%)
because concentrations were not near the limit of quantitation for these analytes. The low check standard
accuracy for ammonium is likely due to the similar issues discussed above for the amines.
Previous IC method precisions reported for use in quantifying the six atmospheric methyl and ethyl amines range
from 0.4 to 17.2 %, which are comparable to our method (Table 2; Chang et al., 2003; Dawson et al., 2014;
Erupe et al., 2010; Huang et al., 2014; Li et al., 2009; VandenBoer et al., 2012; Verriele et al., 2012;). Our
method shows greater average variability than other methods due to our numerous assessments (n = 9) over



multiple months, a more comprehensive analysis compared to previous reports. The sensitivity of this method is
also similar to that of all other reported IC methods as the instrumental detection limits are in the picogram
range. Only VandenBoer et al. (2012) and Chang et al. (2003) report lower detection limits and these are likely a
result of a lower background signal from running the IC methods online. Our method does not achieve
instrumental limits of detection as low as those achieved using derivatization methods coupled with GC-MS or
HPLC analysis (Akyüz, 2007; Fournier et al., 2008; Possanzini and Di Palo, 1990). However, multi-step
derivatization methods are prone to losses that must be quantified with internal standards. These losses can lead
to higher overall method detection limits, which is not the case for direct analysis of water-soluble samples.
Derivatization methods are also difficult to employ for near-real-time analyses of the atmosphere, making the
approach less analytically attractive. Further, our IC method is able to address additional matrix effects that may
result from other atmospheric species by using a sample pre-concentration column. Only positively charged
species are retained in this pre-concentration step and injected through the IC system for analysis, negating
matrix effects from non-charged and anion species.
Employing a method that is capable of quantifying amines at these very low mixing ratios is valuable since
recent work has shown that ppqv concentrations of gaseous amines can lead to particle growth (Almeida et al.,
2013). If our method were applied to online atmospheric ambient sampling of gases or particles the method
could be used to detect amines at sub-pptv mixing ratios. For example, a detectable signal for 100 ppqv mixing
ratios could be attained by sampling through a bubbler, filter, or denuder at a low flow rate of 3 L min$^{-1}$ for 1 –
10 hours, depending on the amine. Sampling may be possible to shorten to an hourly timescale to detect sub-pptv
mixing ratios of atmospheric amines if the method is interfaced with a high sensitivity MS detector. Verriele et
al. (2014) observed a 5 – 30 fold improvement in method detection limits when interfacing their IC method with
a MS detector.
**3.1.3    Expanded amine catalogue for other common atmospheric species**
The separation method developed was further investigated to elucidate its utility in quantifying
monopropylamine (MPA), isomonopropylamine (iMPA) and monobutylamine (MBA), three additional amines
that have been frequently detected in ambient air (Ge et al., 2011). In particular, this test was performed to assess
their potential coelution with the fully separated methyl and ethyl amines. Without modification of the gradient
method, we observed separation of these three additional amines from the original twelve cations with $R_s \geq 0.85$
(Fig. 3a). MPAH$^+$ and iMPAH$^+$ eluted between DMAH$^+$ and TMAH$^+$ and MBAH$^+$ eluted later, but before
TEAH$^+$. The resolution is sufficient between all analyte peaks to allow quantitative analysis of the nine alkyl
amine cations and six inorganic cations. The separation statistics for these additional amines are also presented in
Table 1. Since the additional amines were injected after column degradation had occurred and retention times
had noticeably shifted (see Fig. 3a vs. Fig. 1b), retention time and peak width were estimated using changes in
separation parameters from the original method development for the methyl and ethyl amines. Peak widths for
the propyl and butyl amines were assumed to have increased by approximately 50 %, consistent for the same
increases observed for the methyl and ethyl amines as a result of the column degradation. Retention times for
MPAH$^+$, iMPAH$^+$, and MBAH$^+$ were back-calculated to reflect the initial column conditions using these
corrected peak widths and the resolution values determined from the chromatogram presented in Fig. 3. The





method precisions for iMPAH$^+$, MPAH$^+$ and MBAH$^+$ determined from two standard calibration injections
ranged from 1.3 – 11.9 %. The reported average LODs for both propyl amines are the lowest of the alkyl amines,
while MBAH$^+$ has the highest method LOD because it elutes in a region with a high background due to the step
change in eluent composition not being completely suppressed. The method accuracies for the three additional
amines assessed by both the high and low check standards were within 80 % for all analytes. However, the large
standard deviations in the accuracies for all low check standards highlights the challenge of method
reproducibility for these analytes near the limit of quantitation.

### 3.1.4    Method development with the addition of an inline CG15 guard column

As mentioned previously, IC methods in the literature have been unable to separate potassium from the methyl
and ethyl amines (Huang et al., 2014; VandenBoer et al., 2012) and in our current method potassium has slight
overlap with MMAH$^+$ ($R_S$ = 1.09). We attempted to reduce peak overlap between potassium and the alkyl
amines by adding a crown ether-functionalized CG15 guard column, which has increased selectivity for
potassium, after the CG19/CS19 columns. The addition of the CG15 column resulted in an increased retention
time for potassium of 13 minutes, as expected by using a stationary phase with higher selectivity. The best
separation achieved using the additional guard column is shown in Fig. 3b, where K$^+$ still elutes within the alkyl
amine retention region. The gradient method used to achieve the separation used a flow rate of 1 ml min$^{-1}$, a
column temperature of 55 °C, and held a 1 mM MSA concentration for the first 30 minutes. The eluent
concentration was step increased to 4 mM followed immediately by an exponential ramp to 10 mM over 20
minutes (Chromeleon curve factor = 7). The final concentration of 10 mM is held for an additional 15 minutes,
yielding a total run time of 65 minutes. Even when holding the initial MSA concentration at 1 mM for 50
minutes, the separation was unable to fully resolve the alkyl amine peaks. An increase in retentivity for K$^+$ and
NH$_4^+$ as well as many of the alkyl amines, indicated that the crown ether functionality was not selective for K$^+$ in
this suite of analytes. With the addition of an organic modifier to the mobile phase or the ability to decrease
column temperature, this increase in selectivity from the CG15 column might be harnessed to produce a better
separation. However, due to the limitations of the ICS-2100 system using RFIC we were unable to investigate
these parameters. Furthermore, although a passable separation may be achieved when using a run time greater
than 60 minutes, this would not be as applicable to online analyses as the CG/CS19 method developed without
the addition of the CG15 column. A stationary phase similar to that of the CG/CS19 columns, but with some of
this crown ether selectivity could potentially yield better results than those presented here for the analysis of
atmospheric samples containing large quantities of K$^+$ and amines, such as biomass burning particulate samples.

### 3.1.5    Analytical column stability

Over the course of five months, peak retention times noticeably decreased for all analytes. This is consistent with
what has been previously reported in the literature when hundreds to thousands of injections have been run
through an IC column (VandenBoer et al., 2012). However, this may also be a result of column degradation from
operating the CS19 column at a temperature higher than that recommended by the manufacturers. If this is the
case, in order to prolong column life it is recommended to run the gradient method at a temperature of 30 °C
when conducting offline analysis of samples. If there is an indication that DEAH$^+$ or TMAH$^+$ are present in the
analyzed sample, the column temperature can be increased to separate and quantify the two species. The column





temperature may also be increased to better resolve $DMAH^+$ with $K^+$ if high concentrations of either species are
present.
During the course of method development peak loss of magnesium and calcium was also observed. Cleanup and
eventual replacement of all IC components one by one was not able to resolve this issue. Since retention of
magnesium and calcium on this column or others (e.g. CS12A) never returned, method statistics were calculated
for the two cations using the limited calibrations and chromatograms prior to the occurrence of this system issue.
Accuracy for both cations was unable to be assessed before peak loss occurred. Though loss of magnesium and
calcium during the investigation was unfortunate and unexplained, it did not influence the separation of the
analytes of interest, and the method was still successfully applied to real samples after the loss had occurred.
**3.2      Biomass-burning particle analysis and discussion**
**3.2.1      Size-resolved alkyl amines in particles from an aged biomass-burning plume**
Biomass-burning particles often contain a complex mixture of water-soluble ions, organics, elemental carbon
and other insoluble components, making them nonpareil for testing the robustness of an atmospheric
measurement technique. Ions such as $NH_4^+$ and $K^+$ are consistently detected in biomass burning plumes,
regardless of sampling location as they are well characterized as being co-emitted species (Capes et al., 2008;
Hudson et al., 2004; Pósfai et al., 2003). Particles released during forest fires have also been shown to contain
highly oxidized large molecular weight organics (Di Lorenzo and Young, 2016; Saleh et al., 2014). We tested
the robustness of our method on water-extracted aged biomass-burning particle samples collected by a cascade
impactor in St. John's, Canada. An overlaid chromatogram of two different size-resolved particle samples (100 -
180 nm and 320 – 560 nm) shows the presence of $MMAH^+$, $DMAH^+$ and $DEAH^+$ in the aged biomass burning
samples with complete separation from $K^+$, $NH_4^+$ and $Na^+$ (Fig. 4). The maximum mass loadings for $MMAH^+$,
$DMAH^+$ and $DEAH^+$ were found in particles with diameters ($D_p$) 320 – 560 nm and were $11 \pm 3$ ng m$^{-3}$, $208 \pm 4$
ng m$^{-3}$, $1300 \pm 200$ ng m$^{-3}$, respectively (Table S1). $TMAH^+$, $MEAH^+$ and $TEAH^+$ peaks were also detected in
the BB size-resolved particle extracts. $TMAH^+$ and $TEAH^+$ reached mass loadings of $5 \pm 3$ ng m$^{-2}$ and $4 \pm 2$ ng
m$^{-3}$ respectively, while $MEAH^+$ never exceeded a concentration of 1 ng m$^{-3}$ in any size-resolved particle fraction
(Table S1). Lobert et al (1990) reported detecting $C_1 – C_5$ aliphatic amines from controlled biomass-burning
experiments, which is consistent with our findings. BB derived amines were also identified during the 2007 San
Diego forest fires (Zauscher et al., 2013) and primary amines were observed to make up approximately 6% by
mass of organic content from an aged biomass burning particle sample in British Columbia (Takahama et al.,
2011). However, few studies have addressed the speciation and quantitation of alkyl amines emitted from
biomass-burning events. Schade and Crutzen (1995) estimated the emission rates for MMA, DMA, TMA and
MEA from biomass burning sources based on controlled burn experiments, but do not include a BB emission
rate for DEA or TEA. These emission inventories are yet to include emission rates from atmospheric BB
measurements (Lobert et al., 1990; Schade and Crutzen, 1995).
In Figure 5a we show the molar ratio of the sum of the methyl and ethyl amines to ammonia (which is
considered to be the main atmospheric base), as a function of the size-resolved particles collected. The summed
amine moles exceeded ammonium from 100 to 560 nm, ranged from 0.5 to 1.9 in the fine mode ($PM_1$), with an



average ratio of 0.92 in $PM_1$ calculated using nanoMOUDI bins up to this nominal cutoff. Quantities of $NH_4^+$
were below the detection limit, above 1 µm, yielding no values for the ratio. The large error bars in the ratios are
driven by the low molar quantities of ammonium in the samples as well as a higher than normal variability in the
method blank error on the day of analysis. For these reasons this method blank error was assigned to the size-
resolved samples in place of the $NH_4^+$ error driven by the method precision and accuracy detailed in Table 1.  To
our knowledge, this is the first time that an amines to ammonium ratio greater than one has been reported in any
size resolved fraction of atmospheric particles. An amines to ammonium ratio of approximately 0.05 was
reported by Gibb et al. (1999b) but all other reported ratios have been below 0.05 (Ge et al., 2011). These high
ratios we observed can be attributed to large quantities of $DEAH^+$ and $DMAH^+$. $MMAH^+$ was found to be in
molar quantities 100 times less than that of ammonium while $TMAH^+$, $MEAH^+$ and $TEAH^+$ were found to be in
molar quantities 1000 to 10000 times less than ammonium. Detecting such large molar ratio quantities of
$DEAH^+$ and $DMAH^+$ to $NH_4^+$ in any particle sample is also unprecedented. Mono-substituted amines are the
most frequently detected alkyl amines in atmospheric particles and at molar ratios to ammonium of 1:100 or
lower (Ge et al., 2011; Gorzelska et al., 1990; Mader, 2004, Müller et al., 2009; Yang et al., 2005; Yang et al.,
2004; Zhang et al., 2003). In most instances where di-substituted or tri-substituted amines have been identified in
the particle phase, they are present at molar quantities equal to or less than mono-substituted amines (Healy et
al., 2015; Suzuki et al., 2001). Thus, such high quantities of $DMAH^+$ and $DEAH^+$ in these samples were
unexpected and highly unusual compared to prior reports. In this case, the observation may be due to the fuel
source of the fire or the interaction of the plume with a potent source of atmospheric amines. Previous work has
identified di-substituted amines in large quantities from feedlot plumes (Sorooshian et al., 2008) and in marine
particles (Facchini et al., 2008; Gibb et al., 1999a; Müller et al., 2009; Sorooshian et al., 2009, Youn et al, 2015).
In fact, DMA and DEA have been reported as the second- and third-most abundant organic species in marine
fine aerosol behind methanesulfonic acid during periods of high biological activity in the North Atlantic
(Facchini et al., 2008). Other researchers have also suggested a moderate to high correlation between high
biological activity and di-substituted amine particle mass loadings (Müller et al., 2009; Sorooshian et al., 2009).
Laboratory investigations have shown that methylamines can be produced by marine phytoplankton degradation
of quarternary amines to maintain an osmotic gradient as well as during periods of known zooplankton grazing
(Gibb et al., 1999b). Based on the HYSPLIT back-trajectories calculated for these samples (Di Lorenzo, 2016), it
is possible that the BB plume particles interacted with gaseous DMA and DEA emitted from offshore and
coastal phytoplankton blooms or with enhanced amine emissions in the coastal zone as observed in the marine
boundary layer of California (Youn et al., 2015). The high concentrations of DMA and DEA produced by marine
biological activity could then partition into the biomass burning particles and react to neutralize inorganic acids
(e.g. sulfuric acid), form salts or amides with organic acids, or react with carbonyl moieties in the highly
oxidized organic material produced via BB to form imines (Qiu and Zhang, 2013). This marine amines
hypothesis, while consistent with observations in the literature, is beyond the scope of this work in terms of
assigning the DMA and DEA source.
**3.2.2    Time series of amines in fresh biomass burning plume particles from British Columbia**
Our method was also applied to a time series of $PM_{2.5}$ samples collected at two different locations (BKP and
NVSN) during a forest fire in the Lower Fraser Valley in British Columbia in the summer of 2015. These $PM_{2.5}$





samples were collected every 8 hours while the plume was traversing each site and the $PM_{2.5}$ concentration was
in excess of 200 μg m$^{-3}$. The relative ages for the smoke plumes sampled at both sites were calculated to be 20
hours old or less and back trajectories indicated that the plume did not travel over the open ocean before being
sampled (Di Lorenzo et al., 2016). In these test samples, the method was again able to detect the presence of four
different amines ($iMPAH^+$, $TMAH^+$, $DEAH^+$, and $TEAH^+$) with loadings of $Na^+$, $NH_4^+$ and $K^+$ at ratios in excess
of 100:1. Furthermore, the method was not only able to determine the presence of isomonopropylamine, but also
differentiate it from monopropylamine and trimethylamine, its two structural isomers. $iMPAH^+$, $TMAH^+$,
$DEAH^+$, and $TEAH^+$ had maximum mass loadings in these fresher biomass-burning samples of $60 \pm 40$ ng m$^{-3}$, 9
$\pm$ 7 ng m$^{-3}$, $1.6 \pm 0.8$ ng m$^{-3}$,  and $0.2 \pm 0.1$ ng m$^{-3}$ respectively (Table S2). $iMPAH^+$ was the amine detected in
the largest molar quantities at both sampling sites in British Columbia. The detection of $iMPAH^+$ has not
previously been reported in biomass-burning emission inventories, and based on our measurements may be
important to quantify in future controlled burn experiments. Our results differ from the study conducted by
Takahama et al (2011) on the 2009 forest fires in British Columbia that reports the detection of primary amine
groups, which further suggest that amine emissions from biomass burning and/or their incorporation into
biomass burning particles are not well understood. Although our observed suite of amines includes $iMPAH^+$,
there was no indication of other primary amines from the analyses of the BB particles.
A time-series of the amines to ammonium molar ratio as the smoke plume intrudes into both the BKP and NVSN
sites is presented in Fig. 5b. There were either no amines present or they were present in concentrations below
our detection limits in the ambient particles collected on the front edge of the plume intrusions. When the
maximum $PM_{2.5}$ mass loading of the plume reached the sampling site at t = 0, we saw an absolute maxima in
total amine concentration as well as a relative maxima in the particulate amine to ammonia molar ratio (Fig. 5b).
The particulate amine concentrations and the amines to ammonia ratio then tapered off as the plume diluted and
passed through the site. The measured amines to ammonia ratio in these samples is consistent with previously
reported literature values (Ge et al., 2011). The measured amine species and quantities, as in the aged plume,
could be indicative of the biomass-burning source fuel, fire type, or amine levels in air masses intercepted that
were subsequently incorporated by partitioning and reacting into the condensed phase. Since the smoke plumes
were calculated to be 20 hours old or less and back trajectories indicated that the plume did not travel over the
open ocean, it is less likely that offshore marine amine emissions interacted with the plume. However, the BKP
and NVSN sampling sites are positioned in a coastal urban center and anthropogenic amine emissions from
industry or animal husbandry operations nearby, as well as coastal amine emissions, may still have been
incorporated into the plume before it was sampled.
**4.   Conclusions**
We developed an ion chromatographic method that can separate and quantify nine dominant atmospheric alkyl
amines from common inorganic atmospheric cations. Ion chromatography methods reported in the literature
cannot fully resolve alkyl amine peaks, nor separate interferences from potassium, magnesium and calcium. In
this work, we report the ability to overcome these prevalent issues for atmospheric sampling with a rapidity that
can also be applied to near real-time analyses of aqueous atmospheric extracts by IC. Additionally, the method is
able to separate and quantify three pairs of structural isomers, a limitation for direct particle and gas sampling





mass spectrometry instrumentation in quantifying atmospheric alkyl amines. The method detection limits are
comparable to other published IC methods in the literature, however the described method is not as sensitive as
derivatization methods coupled with GC-MS or LC-MS.
The method is robust. Two sets of BB particle samples collected at two different locations in Canada were
injected onto the IC column and the method detected and quantified amines with the presence of a complex
matrix where inorganic analytes, such as $K^+$, reached ratios of 1000:1 relative to the alkyl amines. This is a
major improvement over all prior reports of the application of IC to the detection of amines in aqueous extracts
of atmospheric particulate matter. Our results suggest that increasing focus on speciation and quantitation of
various alkyl amines from direct BB emissions and their propensity to undergo reactive uptake with biomass
burning particles are needed to constrain global budgets of atmospheric sources and fate of alkyl amines.
Overall, the developed IC method shows promise for i) adoption into standard analysis of water soluble
atmospheric extracts; ii) incorporation into online instrumentation already using ion chromatography for near
real-time analysis of water soluble atmospheric samples; and iii) interfacing with high-resolution mass
spectrometry for even higher analytical sensitivity, particularly where supporting measurements for ppqv-levels
of amines may be stimulating new particle formation in the atmosphere.
*Author contribution.*
TCV designed the experiments and BKP, ATQ, and RAD carried them out. BKP prepared the manuscript with
contributions from all co-authors.
*Competing interests.*
The authors declare that they have no conflict of interest
*Acknowledgements.*
The authors thank Geoff Doerksen at the Lower Fraser Valley Air Quality Monitoring Network in British
Columbia for supplying biomass burning samples. Thanks to Joseph Bautista for help collecting biomass
burning samples in St. John's, as well as Jamie Warren and Kathryn Dawe for their assistance in method
development. Finally, the authors would like to thank Dr. Cora Young for her helpful comments in the writing of
the manuscript. TCV was supported by a Government of Canada Banting Postdoctoral Fellowship and the ICS-
2100 was procured through the Canadian Foundation for Innovation. Funding for this work was provided by the
Government of Newfoundland and Labrador Department of Forestry and Agrifoods through a Centre for
Forestry Science and Innovation grant (Project No. 221269).





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



1    **Figures**

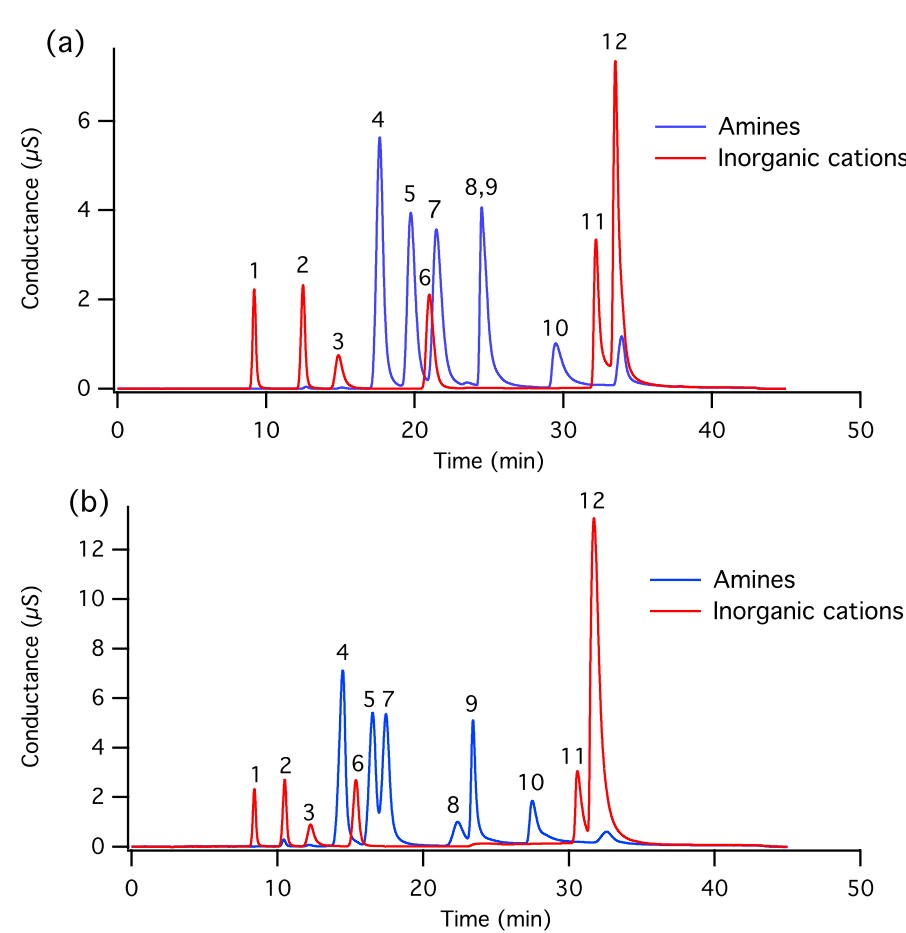

Figure 1. Separation of amine and inorganic cation standards with the highest resolution gradient program at (a)
30 °C and (b) 55 °C. The order of elution in (a) is as follows: $Li^+$ (1), $Na^+$ (2), $NH_4^+$ (3), $MMAH^+$ (4), $MEAH^+$
(5), $K^+$ (6), $DMAH^+$ (7), $TMAH^+$ (8), $DEAH^+$ (9), $TEAH^+$ (10), $Mg^{2+}$ (11), and $Ca^{2+}$ (12). Cation peaks are
labeled according to the same identities in (b).





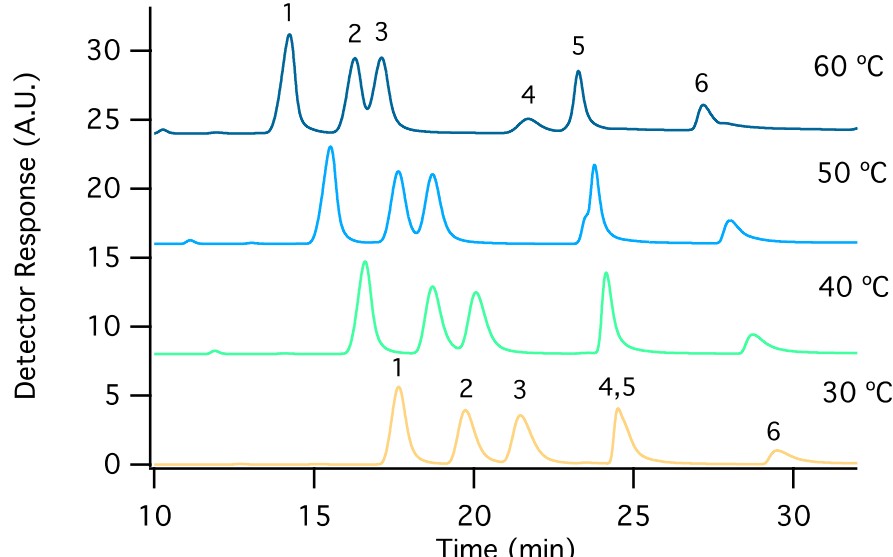

Figure 2. Separation of 1 μg ml$^{-1}$ mixed amines standard with the final method gradient elution program at 30
°C, 40 °C, 50 °C and 60 °C. The peak elution order was MMAH$^+$ (1), MEAH$^+$ (2), DMAH$^+$ (3), TMAH$^+$ (4),
DEAH$^+$ (5), and TEAH$^+$ (6). The separation of diethylamine (DEA) from trimethylamine (TEA) was achieved at
column temperatures greater than 50 °C.



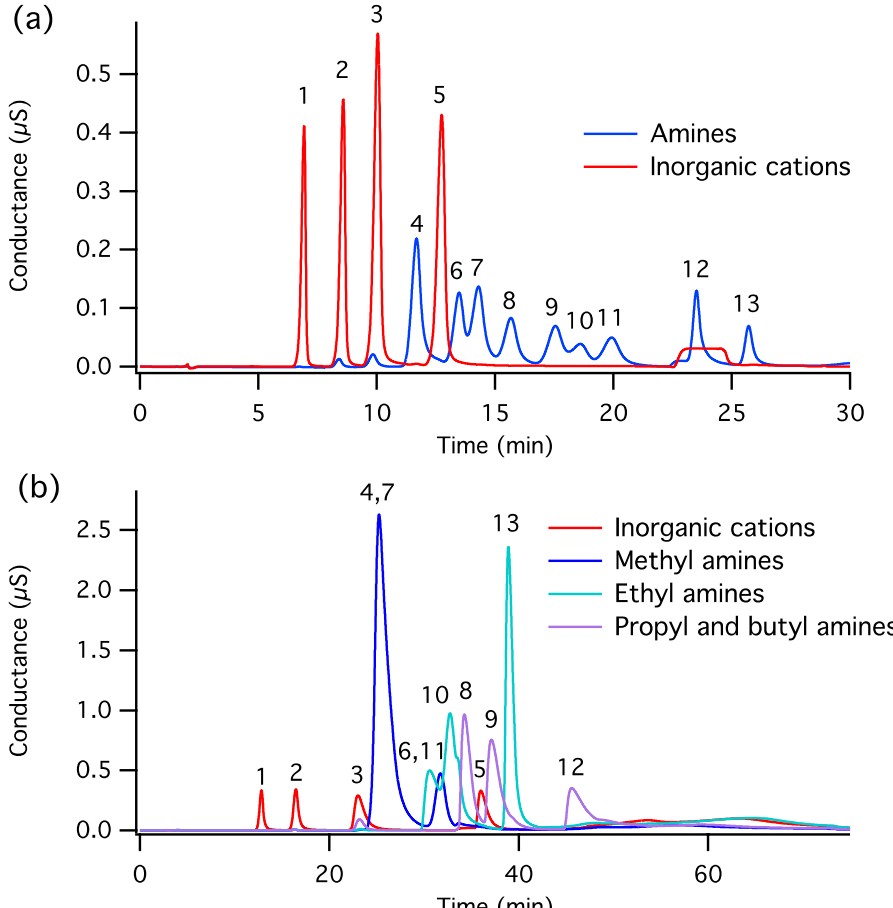

Figure 3. (a) Separation of amine and inorganic cation standards with the addition of MPAH+, iMPAH+ and
MBAH+ using the final gradient program. The order of elution in (A) is as follows: Li+ (1), Na+ (2), NH$_4$+ (3),
MMAH+ (4), K+ (5), MEAH+ (6), DMAH+ (7), iMPAH+ (8), MPAH+ (9), TMAH+ (10), DEAH+ (11), MBAH+
(12), and TEAH+ (13). (b) Separation of amine and inorganic cation standards with the addition of MPA, iMPA
and MBA and the addition of the CG15 column using a modified gradient program. Cation peaks are labeled
according to the same identities in (b).





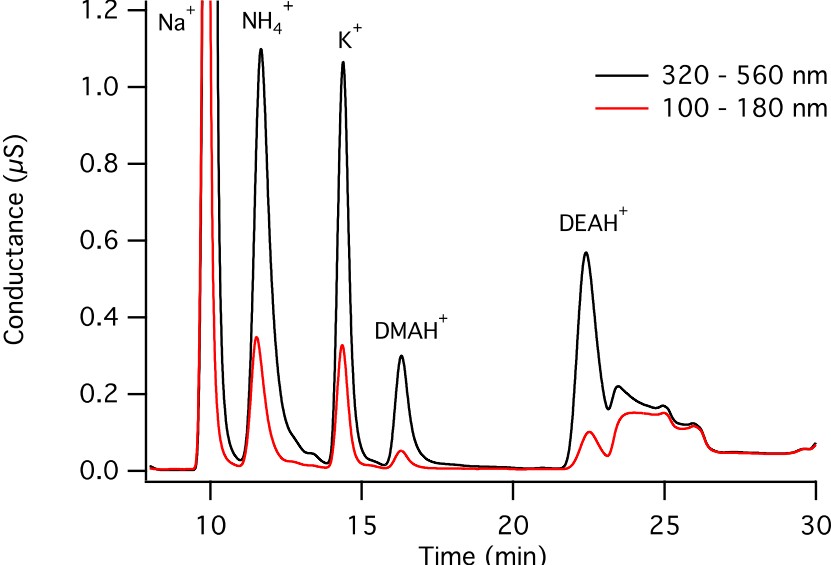

Figure 4. Overlaid chromatograms of MOUDI size-fractionated particle samples collected in St John's on July 6,
2013 during the intrusion of a biomass-burning plume that originated from Northern Labrador and Quebec. The
robustness of the separation method for MMAH[+], DMAH[+] and DEAH[+] from the common inorganic cations is
demonstrated for the 320-560 nm (Black) and 100-180 nm (Red) size bins.



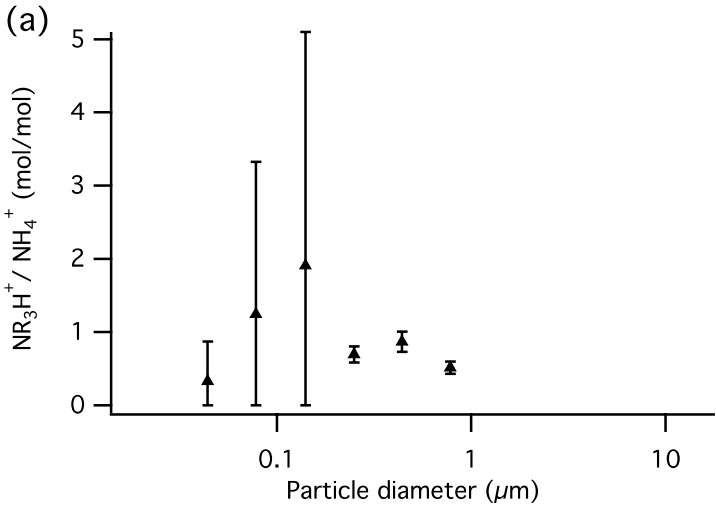

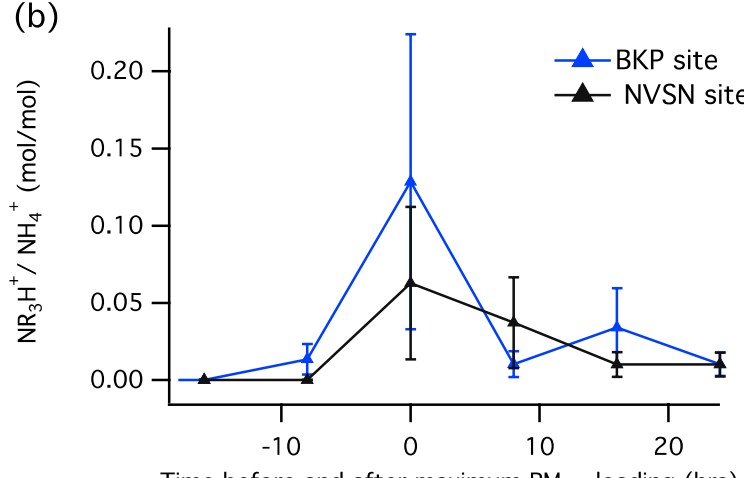

Figure 5. (a) Amines to ammonium ratio in the size-resolved aged biomass-burning sample originating from Quebec and Labrador in the summer of 2013. (b) Amines to ammonium ratio for the Burnaby/Kensington Park (BKP) site and North Vancouver/Second Narrows (NVSN) site in British Columbia during the summer 2015 wildfires.



**Tables**
Table 1. Separation characteristics and statistics for the CS19 gradient method. Retention times ($t_r$), peak width and resolution were determined using the highest
calibration standard for the alkyl amine (500 ng) and inorganic (160 – 520 ng) cations. Sensitivity, precision, average LOD, and LOD range were analyzed using multiple
calibration standards and blanks (see Section 2.4). Upper and lower range accuracies were assessed using six high and low check standards (n = 6) for the alkyl amine
cations and four high and low check standards (n = 4) for the inorganic cations. The low check standards were 15 times more concentrated than the lowest standard and
the high check standards were 150 times more concentrated than the lowest standard.

| Cation | $t_r$ (min) | Peak width (min) | Resolution | Sensitivity ($\mu S{*}min$ $mol^{-1}$) | Precision ($\pm$ $1\sigma$) (%) | Upper range accuracy (%) | Lower range accuracy (%) | Average LOD ($\pm$ $1\sigma$) (pg) | LOD range (pg) |
|---|---|---|---|---|---|---|---|---|---|
| $Li^+$ | 8.56 | 0.72 | 2.68 | 11.5E08 | 2 | 96 $\pm$ 5 | 82 $\pm$ 4 | 0.6 $\pm$ 0.2 | 0.3 - 0.8 |
| $Na^+$ | 10.50 | 0.73 | 1.87 | 5.04E08 | 2 | 95 $\pm$ 4 | 90 $\pm$ 6 | 8 $\pm$ 4 | 4 - 14 |
| $NH_4^+$ | 12.28 | 1.18 | 1.85 | 2.45E08 | 4 | 103 $\pm$ 4 | 50 $\pm$ 50 | 22 $\pm$ 17 | 7 - 47 |
| $MMAH^+$ | 14.17 | 0.86 | 1.09 | 1.42E08 | 5 | 98 $\pm$ 6 | 40 $\pm$ 30 | 300 $\pm$ 300 | 30 - 650 |
| $K^+$ | 15.11 | 0.87 | 1.22 | 4.14E08 | 5 | 99 $\pm$ 2 | 94 $\pm$ 7 | 14 $\pm$ 11 | 4 - 28 |
| $MEAH^+$ | 16.12 | 0.79 | 1.08 | 0.90E08 | 7 | 97 $\pm$ 5 | 40 $\pm$ 10 | 500 $\pm$ 200 | 200 - 700 |
| $DMAH^+$ | 17.01 | 0.85 | 1.64 | 1.48E08 | 5 | 100 $\pm$ 10 | 30 $\pm$ 30 | 200 $\pm$ 300 | 40 - 650 |
| $iMPAH^+$ | 18.35 | 0.79 | 2.24 | 0.84E08 | 4 | 90 $\pm$ 10 | 80 $\pm$ 80 | 70 $\pm$ 40 | 40 - 90 |
| $MPAH^+$ | 20.22 | 0.88 | 1.55 | 0.62E08 | 12 | 88 $\pm$ 4 | 90 $\pm$ 90 | 50 $\pm$ 40 | 20 - 80 |
| $TMAH^+$ | 21.76 | 1.11 | 1.48 | 0.34E08 | 16 | 90 $\pm$ 10 | 30 $\pm$ 20 | 600 $\pm$ 300 | 300 - 1200 |
| $DEAH^+$ | 23.45 | 1.18 | 2.51 | 0.76E08 | 8 | 97 $\pm$ 8 | 50 $\pm$ 30 | 400 $\pm$ 300 | 100 - 800 |
| $MBAH^+$ | 25.47 | 0.43 | 3.12 | 0.62E08 | 1 | 80 $\pm$ 20 | 100 $\pm$ 80 | 910 $\pm$ 30 | 890 - 930 |
| $TEAH^+$ | 27.62 | 0.95 | 3.40 | 0.85E08 | 12 | 96 $\pm$ 4 | 49 $\pm$ 6 | 800 $\pm$ 400 | 500 - 1400 |
| $Mg^{2+}$ | 30.59 | 0.79 | 1.22 | 12.2E08 | 1 | --- | --- | 2000 $\pm$ 3000 | 200 - 4000 |
| $Ca^{2+}$ | 31.72 | 1.05 | N/A | 14.3E08 | 2 | --- | --- | 3700 $\pm$ 200 | 3500 - 3800 |





1  Table 2. Analytical performance of other IC methods used for the determination of atmospheric methyl and ethyl
2  amines by conductivity detection (CD) or mass spectrometry (MS).

| Analyte | Detection method | Pre-conc | Column | LOD (pg) | Precision (%) | Reference |
|---|---|---|---|---|---|---|
| MMAH⁺ | CD | Yes | CS10 | 31 | 2 – 2.7 | Chang et al., 2003 |
| | CD | Yes | CS12A | 18 | 4.5 | VandenBoer et al., 2012 |
| | CD | No | CS14 | 2500 | 3.8 | Verriele et al., 2012 |
| | MS | No | CS14 | 500 | 5.8 | Verriele et al., 2012 |
| | CD | Yes | CS17 | 540 | 4.8 | VandenBoer et al., 2012 |
| | CD | Yes | CS19 | 30 - 650 | 5 | This work |
| | CD | No | Metrosep C2 | 21500 | 0.4 | Erupe et al., 2010 |
| | CD | Yes | Metrosep C4 | 2100 | 12.2 | Huang et al., 2014 |
| | CD | No | Metrosep C4 | 160 | 7.3 | Dawson et al., 2014 |
| DMAH⁺ | CD | Yes | CS10 | 40 | 2 – 2.7 | Chang et al., 2003 |
| | CD | Yes | CS12A | 25 | 1 | VandenBoer et al., 2012 |
| | CD | No | CS14 | 4000 | 10.5 | Verriele et al., 2012 |
| | MS | No | CS14 | 150 | 11.4 | Verriele et al., 2012 |
| | CD | Yes | CS17 | 870 | 14 | VandenBoer et al., 2012 |
| | CD | No | CS17 | 1500 | 1.2 | Li et al., 2009 |
| | CD | Yes | CS19 | 40 - 650 | 5 | This work |
| | CD | No | Metrosep C2 | 23000 | 1.4 | Erupe et al., 2010 |
| | CD | Yes | Metrosep C4 | 3800 | 15.7 | Huang et al., 2014 |
| | CD | No | Metrosep C4 | 320 | 1.1 | Dawson et al., 2014 |
| TMAH⁺ | CD | Yes | CS10 | 26 | 2 – 2.7 | Chang et al., 2003 |
| | CD | Yes | CS12A | 220 | 1 | VandenBoer et al., 2012 |
| | CD | No | CS14 | 2500 | N/A | Verriele et al., 2012 |
| | MS | No | CS14 | 500 | 12.2 | Verriele et al., 2012 |
| | CD | Yes | CS17 | 1580 | 3.3 | VandenBoer et al., 2012 |
| | CD | No | CS17 | 2000 | 3.5 | Li et al., 2009 |
| | CD | Yes | CS19 | 300 - 1200 | 16 | This work |
| | CD | No | Metrosep C2 | 38000 | 1.1 | Erupe et al., 2010 |
| | CD | No | Metrosep C4 | 970 | 6.1 | Dawson et al., 2014 |
| MEAH⁺ | CD | Yes | CS10 | 37 | 2 – 2.7 | Chang et al., 2003 |
| | CD | Yes | CS12A | 33 | 12 | VandenBoer et al., 2012 |
| | CD | No | CS14 | 1000 | 5.1 | Verriele et al., 2012 |
| | MS | No | CS14 | 500 | 7.9 | Verriele et al., 2012 |
| | CD | Yes | CS17 | 790 | 10 | VandenBoer et al., 2012 |
| | CD | Yes | CS19 | 200 - 700 | 7 | This work |



| | | | | | | |
|---|---|---|---|---|---|---|
| | CD | Yes | Metrosep C4 | 2200 | 4.3 | Huang et al., 2014 |
| DEAH⁺ | CD | Yes | CS12A | 195 | 14 | VandenBoer et al., 2012 |
| | CD | No | CS14 | N/A | N/A | Verriele et al., 2012 |
| | MS | No | CS14 | 35 | 9 | Verriele et al., 2012 |
| | CD | Yes | CS17 | 1140 | 3.5 | VandenBoer et al., 2012 |
| | CD | Yes | CS19 | 100 - 800 | 8 | This work |
| | CD | Yes | Metrosep C4 | 4100 | 4.6 | Huang et al., 2014 |
| TEAH⁺ | CD | Yes | CS12A | 32000 | 2 | VandenBoer et al., 2012 |
| | CD | Yes | CS17 | 1870 | 5.9 | VandenBoer et al., 2012 |
| | CD | Yes | CS19 | 500 - 1400 | 12 | This work |
| | CD | Yes | Metrosep C4 | 15900 | 5.1 | Huang et al., 2014 |

