# Peer review of "Quantitation of nine alkyl amines in atmospheric samples"

_Atmospheric Measurement Techniques, 2016_

## Referee Comment (RC1) · Anonymous Referee #2 · 7 Dec 2016

The paper presents a detailed development of an analytical method for alkyl amines in aerosol particles based on ion chromatography. After a detailed optimization of several parameters, the method was applied to some field samples from biomass burning events. The topic is of high importance; selective and sensitive well-characterized analytical methods are strongly needed in atmospheric measurements. The paper is well written however, there are some points that need clarification before publishing:

Page 3, line 8: The authors point out, that the matrix of atmospheric samples can cause problems in the analytical measurements (which is true). Still I think that the problem of matrix effects is not too much discussed in the manuscript concerning this analytical method. For example on page 10, line 10/11 the authors write that using a

pre-concentration column, you can eliminate negative charged and neutral disturbing ions. Did you test this here? And still the separation of the amines from other dominating cations that are for example present in marine samples in high quantities (sodium, ammonium) would be a problem – they would be enriched using a pre-column. This could affect again the separation of the (lower concentrated) amines

Generally, I was wondering if you performed some further tests regarding matrix effects. Did you for example perform standard addition (to prove the accuracy of the method) with the field samples? This would be a high value in the method validation.

Page 6, Chapter 2.5 and 2.6. The authors use different types of field samples for the measurements of the amines with their method: nano Moudi aluminium foils and PM 2.5 filter samples. I was wondering if you considered measurements of field blanks, e.g. did the filter material and the sampling technique affect the sampling of the amines? Did you observe artifacts (e.g. from the gas phase) using the filter samples? This topic was addressed for example in Müller et al. (2009) in ACP that the authors cite here. Could you comment on this topic?

Page 8, line 13 and following: The authors report about the temperature effect in the IC separation. Also other authors have used temperatures above 50 °C e.g. van Pinxteren et al., (J Atmos Chem (2015) 72: 105, doi:10.1007/s10874-015-9307-3), have used a column temperature of 60°C in the IC separation of aliphatic amines. Maybe you can include previous publications that regard temperature effects in IC separation.

Page 8, line 32: ". . . have detected large quantities of MEA and DEA in ambient air. . ." do you refer to gas phase or aerosol measurements in the cited literature? Please specify.

Page 10, chapter 3.1.3 / Page 11, chapter 3.1.5: The authors discuss the problem of decreasing separation due to column instability. The authors mention that the high column temperature might accelerate this process. I am however sceptic if it is a good and practical solution to measure all samples at 30 °C and to repeat all measurements

that indicate the presence of DEA or TEA at higher temperatures. Do you have any marker for column stability that indicate when you should replace the column? As the column degradation occurs gradually you might lose peak information in your samples quite soon (especially regarding field samples).

Page 12: line 3 and following: The loss of calcium and magnesium. Could you speculate about the loss? Is it irreversible bound to the column surface? In the conclusion, you describe that ion chromatography methods often suffer from interferences with magnesium or calcium and the present work overcomes this problem. However, it can not be the solution to have an "unfortunate and unexplained" loss of these elements. Did you consult the manufactures about this problem? What is your future strategy here?

Page 12, line 37: unclear phase; did you mean "... in the size range of 100 to 560 nm"? Please clarify.

Page 13, line 21 and following: The high ratio of amines to ammonium is very surprising. As one explanation, the authors state that marine influences could be an explanation for this observation. While amines can indeed have significant marine sources (as also stated by the authors), also ammonium would be elevated then (as also shown in the cited literature, e.g. Gibb et al). So do you think that there could be a marine source producing so much more amines than ammonium? Is there any evidence in the literature?

Page 13, line 22: The authors might consider including the recent publication of van Pinxteren et al. 2015, as they reported high concentration of the amines in a marine region close to the Cape Verdean islands.

Page 13, line 28-31: Did you measure other marine tracers (like MSA) to test the hypothesis?

---

## Referee Comment (RC2) · Anonymous Referee #1 · 12 Dec 2016

This manuscript presents a method for quantification of atmospheric amines by ion chromatography. Detection and quantification of amines is an important area of research as few studies have focused on understand which amines are present beyond the methylamines. This is important work to publish to aid other in quantifying atmospheric amines. There are a few concerns about the manuscript in its current form. The loss of detection of magnesium and calcium during method testing suggests issues related to the IC method (improper suppressor current for eluent concentration) is concerning. As is the lack of blank filter samples analyzed to address potential issues in background especially since the glassware needed to be specially cleaned. Expanded comments on each of these issues and other areas are below.

[Figure]

Are there shifts in retention time in mixed cation-amine standards? Maybe cases where $NH_4+$ or $K+$ are many times higher than the amine concentration? For your example chromatograms (not real samples) you are showing the amine and cations separately but I wonder if there are changes to retention time with increase cation concentration or differences in the peaks based on the matrix. And is it really appropriate to analyze the standards separately when determining concentrations in complex (real) samples. It seems that applications of this method might be in samples where the amine concentrations are much lower than other cation species. It would be helpful to address potential limitations, if they exist, for the upper end of cation to amine ratios.

Pg 4-5: optimized separation It would be helpful if you want others to use the method you've developed to include the text of the Chromeleon method file in the supplement. You are also missing important information including suppressor current, re-equilibration time, and injection volume. What is the typical pressure in the system?

Pg 5 Line 10-13: How much contamination do you see without the involved cleaning procedure? Do you see amines or just cations? Where could background amines be coming from?

Pg 5 Line 29: What is a calibration blank? The zero point in the calibration curve? Please explain in the manuscript. It might be nice to show an example in the supplement. This would also help to explain the peaks related to changes in the eluent.

Pg 6: BB samples Were blanks analyzed to determine background? Based on your thorough glassware and storage container cleaning procedure it seems like there might be background on your filters due to handling the filtering process. To report atmospheric concentrations without taking into account the filter/procedure blanks is concerning.

Pg 8: 35 Can you really separate Mg and Ca from the amines? In Fig 1 there appears to be a peak in amine trace at the same time Ca elutes. It is unclear to me what this peak is and where MBAH+ comes out relative to Mg and Ca. The loss of the ability to

detect Mg and Ca complicates this but it is an important consideration since Mg and Ca are commonly found in atmospheric aerosols. The change in the ability of detect Mg and Ca is very concerning will make it difficult for others to use your method.

Pg 9: 18 Could some range of variability be related to baseline noise from a failing/failed suppressor? See comment below related to loss of Mg and Ca detection.

Pg 10: 32 How long had the column been used when the degradation occurred? Is this method realistic to use for regular analysis if the column degrades with time. With so many peaks, often at low concentrations, it is critical to know when everything elutes.

Pg 12: 3-9 Loss of Mg and Ca is likely related to the suppressor no longer working as it should. In my experience Mg and Ca are the first to go when a suppressor stops working. Was it replaced with a new one (that worked with a traditional CS12 method)? There could be changes to the sensitivity of other species if the suppressor stopped working properly, especially those with the later retention times. We've observed that if you aren't controlling the current properly for the low concentration of MSA in the eluent you can go through a lot of suppressors. The manual for the SRS300 provides guidance for calculating the correct current setting. Please discuss the suppressor setting and potential implications if it wasn't working as expected.

Pg 13: 4 Please explain what the method blank is.

Comments on Figures:

Please include nominal concentrations of the cations and amines or someway to know if the change in peak height is a function of the method parameters or concentration. For example you suggest comparing Figure 3a and 1b (Pg 10 line 33) to see the shift in retention time - I can't help but notice the y-scales are very different. When you are losing peaks and seeing column degradation it is important to know if the peak height/area is also changing. The statements about the calibration curves suggest the areas didn't change over the course of your study but by including the concentration
information for each chromatogram/injection you would clearly show that.

In Figure 3b I like the separation of the types of amines. I think it would be helpful for the comparison between a and b to do the same for a, if possible. Or some type of color-coding by type.

Figure 5a: add a summation sign in y-axis label to clarify it is the sum of the amines. And add in the caption it is the sum of the methyl and ethyl amines as you state in the text. Also add what the error bars represent.

Figure 5 b: is this also ratio of the sum of the methyl and ethyl amines to ammonium?

Table 2 seems more appropriate for the supplement.

Minor Comments:

Pg 4: 9-11 Please define all amines here so the definitions are easily found in one place.

Pg 7:2 Remove therefore

Pg 10: 15 Please define ppqv

Pg 14: 1-2 Did the PM2.5 concentration need to be in excess of 200 ug/m3 for samples to be collected during the 8 hr period? Please clarify.

---

## Author Comment (AC1) · 1 Feb 2017

The Authors would like to thank both Reviewers for taking the time to provide this constructive feedback on our discussions paper. The Reviewer comments have highlighted some concerns in our original manuscript and have also provided us with suggestions to resolve issues we experienced during our method development. In light of this feedback, we have performed a standard addition experiment on a biomass-burning sample to validate the method in the presence of a complex matrix. This suggestion provided by the Reviewers has improved the manuscript by showcasing the method's robustness and applicability to environmental samples. Further, after receiving valuable experiential input from the Reviewers we were able to regain peak-to-peak resolution

of magnesium and calcium in our method after replacing the CERS 500 suppressor. The troubleshooting of this issue has now allowed us to calculate the accuracies for these two analytes and comment on their overall method performance statistics. This has also allowed us to add separation information for two alkyl diamine species and ethanolamine, all of atmospheric interest, to the manuscript (see Page 12, Lines 21-32). In addition, we have added critical components to our methods section and subsequent discussion to address the Reviewers' concerns. For example, a discussion of calibration and method blanks and a corresponding figure has now been added to the manuscript.

Each comment from both Reviewers has been addressed in detail below. We have provided a response for each comment and have provided the changes in the manuscript as required. The revised manuscript denotes these changes in blue text. Additions or changes to the manuscript have been highlighted using the corresponding page and line numbers in the revised manuscript. Any updated figures can be found in the uploaded manuscript or supporting information. By addressing all of the concerns of the Reviewers we feel that the updated manuscript is significantly improved. Finally, we have also made minor changes throughout the manuscript to increase clarity and build on the excellent commentary of the Reviewers.

Reviewer 1:

Comment 1: Are there shifts in retention time in mixed cation-amine standards? Maybe cases where NH4+ or K+ are many times higher than the amine concentration? For your example chromatograms (not real samples) you are showing the amine and cations separately but I wonder if there are changes to retention time with increase cation concentration or differences in the peaks based on the matrix. And is it really appropriate to analyze the standards separately when determining concentrations in complex (real) samples. It seems that applications of this method might be in samples where the amine concentrations are much lower than other cation species. It would be helpful to address potential limitations, if they exist, for the upper end of cation to amine

ratios.

Response 1: We have addressed most of the Reviewer's concerns above by performing a standard addition of amines to a size-resolved BB sample extract and added a table detailing the results to the supplementary information (Table S1). The sample chosen for our standard addition contained very high Na+, NH4+ and K+ content, which is at the upper end of the ratios near 1000:1 we expect in real samples. In short, the BB sample extract matrix has little effect on retention time, none on instrument sensitivity, and therefore separate external cation calibrations, fully mixed cation calibrations, calibrations with representative analyte cation ratios, and standard additions will provide the same results upon quantitation. We expect this due to the selectivity of the preconcentration column for cations in the sample and exclusion of much of the matrix prior to sample injection on the analytical column. Furthermore, all BB samples were diluted such that all cation analytes were within the calibration range to mitigate any column overloading matrix effects. We have added the following lines to the manuscript to clarify the performance of this method with respect to matrix effects:

Page 6, Lines 9-15:

To assess the method robustness in the presence of a complex matrix the gradient method standard addition was performed on a subsample of a size-resolved BB particle extract (320 – 560 nm) (see Sect. 2.5). Standard addition was performed by adding known quantities of methyl and ethyl amine solution to a 0.5 ml sample of the extract followed by dilution to 5 ml. The amount of the methyl and ethyl amines added to the internal calibration matched that of the external calibration. The slope and retention times for the methyl and ethyl amines from the internal calibration were calculated and compared to those performed externally to quantify matrix effects. Discussion of the analytical performance of the CS19 gradient program is presented in Sect. 3.1.2.

Page 6, Lines 24-29:

A sub-sample of each aluminum substrate (10 % of the total substrate area) was extracted into a glass vial with 5 mL ultrapure deionized water by sonication (VWR Scientific Products/Aquasonic 150HT, Ultrasonic Water Bath) for 40 minutes. The extracts were filtered using a 0.2 $\mu$m polytetrafluoroethylene (PTFE) filter and stored in polypropylene vials at 4 °C prior to analysis by IC within 24 hours. Cation analytes within these samples all fell within their respective calibration ranges and did not require any further dilution.

Page 7, Lines 9-12:

Approximately 37% of each filter spot area was placed into a polypropylene vial with 10 ml of deionized water, and sonicated for 40 minutes. The extracts were filtered with PTFE syringe filters (3 mm diameter, 0.2 $\mu$m pore size, VWR International, Radnor, Pennsylvania, USA) and diluted by a factor of five with ultrapure deionized water so that all analytes were in the calibration range before being injected on the IC.

Page 10, Lines 33-39 and Page 11, Lines 1-5:

To further test the efficacy of the method, a standard addition calibration was performed in the presence of the complex BB matrix. The calibration slopes and retention times for each analyte from the standard addition and external calibration performed on the same day are listed in Table S1. The slopes for the two calibrations varied between 0 and 8 %, which is within the method calibration precisions presented in Table 1. Thus, the BB sample extracts did not exhibit matrix effects as the slopes were found to be within the expected instrument variability as determined by external calibration standard analysis. However, increasing retention times of approximately 0.3 − 0.5 minutes were observed for all cation analytes when performing the standard addition. This is an effect inherent in IC when samples with higher total quantities of cations are preconcentrated, resulting in a sample plug filling a greater volume of the stationary phase capacity. The initial weak mobile phase of the gradient method will therefore take a greater amount of time to elute all of the analyte cations from the preconcentration and analytical columns. This same increase in retention times is present in the external

calibration with increasingly concentrated standards (Table 1).

Comment 2: optimized separation It would be helpful if you want others to use the method you've developed to include the text of the Chromeleon method file in the supplement. You are also missing important information including suppressor current, reequilibration time, and injection volume. What is the typical pressure in the system?

Response 2: The Reviewer is correct in noting that much of this additional information would be of use. The text file for the Chromeleon method file has been noted as available in the manuscript and added to the S.I. (Fig. S1). The following information has been added to the method section:

Page 5, Lines 11-15:

The IC was returned to initial conditions and re-equilibrated for 10 minutes as the next 1 ml sample aliquot was prepared for injection by the AS-DV. The suppressor current, optimized for this flow rate and the maximum eluent concentration in accordance with the calculation provided by the manufacturer, was set at 37 mA. The typical backpressure in the system at these conditions was 2100 psi. The Chromeleon method file for the method described above is detailed in Fig. S1.

Comment 3: How much contamination do you see without the involved cleaning procedure? Do you see amines or just cations? Where could background amines be coming from?

Response 3: The rigorous cleaning procedure used in the manuscript is the standard protocol in our analytical laboratory and guarantees that there is minimal contamination in our standards and samples for all our analytical purposes, not only ion chromatography. The samples presented in this manuscript were analyzed for multiple analytes on additional instrumentation so glass and plasticware were washed accordingly. There were no background amines detected in any of our calibration, reagent and/or field blanks (e.g. Fig. S2). However, there was always a small background of sodium,

ammonium, potassium and calcium present in quantities below our lowest cation calibration standard in all analytical blanks. These arise from our ultrapure water system. Somewhat greater quantities of these analytes were detected in method and sampling blanks due to leaching from the holding vessels and/or field collection substrates. In such cases that these inorganic cations were detected in our blanks, sample analyte amounts were corrected by an appropriate subtraction prior to quantitation. The following lines have been added to the manuscript to address the Reviewer's comments:

Page 5, Lines 24-29:

The mass range for each inorganic cation calibration is as follows: $Li+$ (0.82-160 ng), $Na+$ (7.8-160 ng), $NH4+$ (8.4-170 ng), $K+$ (26-520 ng), $Mg2+$ (6.4-130 ng), and $Ca2+$ (18-360 ng). All calibration curves contained trace inorganic cation impurities from the ultrapure deionized water source or holding vessels that fell below the lowest calibration standard, and were corrected accordingly to allow for inter-day method performance comparison. Trace quantities of amines were not observed. An example of a calibration blank chromatogram is presented in Fig. S2.

Comment 4: What is a calibration blank? The zero point in the calibration curve? Please explain in the manuscript. It might be nice to show an example in the supplement. This would also help to explain the peaks related to changes in the eluent.

Response 4: The Reviewer is correct in the assumption that our calibration blank is the zero point in our calibration curve, which is often referred to as the analytical blank or reagent blank. We have clarified this in the manuscript. As mentioned in the previous response, we have added a blank chromatogram to the SI (Fig. S2) and identified the system peak related to the stepwise increase in eluent concentration. The following lines have been modified in the manuscript to describe our calibration method blank:

Page 5, Line 22:

The mixed amine standards and mixed inorganic cation standards were prepared separately and each cation standard set was composed of five calibration standards, two check standards and an ultrapure deionized water blank.

Comment 5: BB samples were blanks analyzed to determine background? Based on your thorough glassware and storage container cleaning procedure it seems like there might be background on your filters due to handling the filtering process. To report atmospheric concentrations without taking into account the filter/procedure blanks is concerning.

Response 5: We would like to thank the Reviewer for their concern. This was not clearly stated in the manuscript, despite field blank corrections having been appropriately applied in the calculations of the concentrations of the samples we have reported. We carried blank filters through the entire collection and extraction process in tandem with the actual samples to account for all systematic handling error, which may have resulted in contamination of the samples. All BB samples were blank-corrected with a representative field blank using the procedures detailed below. Error arising from this process was always taken into account. We would also like to reiterate that there were no amines in any of the field blanks analyzed, only measurable quantities of inorganic cations. To further address this concern we have added chromatographic traces for field sampling blanks to the analytical blank trace depicted in the SI in Fig. S2.

Page 6, Lines 22-34:

The nanoMOUDI samples were collected on 13 aluminum substrate stages in to size-resolved bins of atmospheric particles with a diameter range spanning 0.010 –18 $\mu$m. Air was sampled continuously for 25.5 hours at a flow rate of 30 L min-1. A sub-sample of each aluminum substrate (10 % of the total substrate area) was extracted into a glass vial with 5 mL ultrapure deionized water by sonication (VWR Scientific Products/Aquasonic 150HT, Ultrasonic Water Bath) for 40 minutes. The extracts were filtered using a 0.2 $\mu$m polytetrafluoroethylene (PTFE) filter and stored in polypropylene vials at 4 °C prior to analysis by IC within 24 hours. Cation analytes within these

samples all fell within their respective calibration ranges and did not require any further dilution. An aluminum substrate field blank was also transported and exposed to the ambient atmosphere briefly at the collection site before being stored in a sealed container for the duration of the sample collection, transported back with the samples, and extracted simultaneously following the same procedure. All calculated quantities were corrected with measurements of the field blank and additional error from this correction propagated into our final reported values. The field blank chromatogram for the size-resolved BB samples is presented in Fig. S2.

Page 7, Lines 8-14:

Particles were collected on glass microfiber filter tape and stored at -20 °C until extracted. Approximately 37% of each filter spot area was placed into a polypropylene vial with 10 ml of deionized water, and sonicated for 40 minutes. The extracts were filtered with PTFE syringe filters (3 mm diameter, 0.2 $\mu$m pore size, VWR International, Radnor, Pennsylvania, USA) and diluted by a factor of five with ultrapure deionized water so that all analytes were in the calibration range before being injected on the IC. During the extraction, an unexposed area of the glass microfiber filter tape was sampled and extracted for use as a field blank. All calculated quantities and errors were blank-corrected using the field blank.

Comment 6: Can you really separate Mg and Ca from the amines? In Fig 1 there appears to be a peak in amine trace at the same time Ca elutes. It is unclear to me what this peak is and where MBAH+ comes out relative to Mg and Ca. The loss of the ability to detect Mg and Ca complicates this but it is an important consideration since Mg and Ca are commonly found in atmospheric aerosols. The change in the ability of detect Mg and Ca is very concerning will make it difficult for others to use your method.

Response 6: These are very good points that the Reviewer has raised, which we did not clearly address in the manuscript. The amine peak that is unlabeled and appears underneath the calcium peak in Figure 1 is in fact trace calcium contamination in the

standard, equivalent to that which was present in our analytical blanks even after our rigorous cleaning procedure and is likely from our deionised water system as an ultratrace contaminant in the nominally 18.2 MΩ water. This is consistent with our past detection of trace cations in a variety of deionised water systems, especially when also utilizing a preconcentration column. We have removed the peak labels and updated the captions on Figures 1 and 3 to improve on this. All sample quantities and standard statistics were corrected for this contamination up until the time where we experienced severe peak broadening and loss in resolution of calcium and magnesium in our method.

The MBAH+ coelution noted by the Reviewer is coincident with a system peak derived from the MSA step change in our gradient method. In the blank chromatogram we have added to the supporting information to help clarify where the system peak elutes, this has been labeled (Fig. S2). MBAH+ does not coelute with Mg2+ or Ca2+ as it elutes before TEAH+ and the divalent inorganic cations are known to elute after TEAH+, as in Figure 1. Although we experienced loss of peak shape, but not total area, for the two divalent cations, we still do not expect the selectivity and relative retention order for this suite of cations to change over time. Our evidence for this is the retention time stability we report for the remainder of our cations.

Comment 7: Could some range of variability be related to baseline noise from a failing/failed suppressor? See comment below related to loss of Mg and Ca detection.

Response 7: A failing suppressor could definitely result in increased baseline noise, however the changes in baseline noise that we quantified for our detection limits were random and did not follow a trend of increasing noise over time. Peak heights and peak areas for all the amines and inorganic cations (besides calcium and magnesium) also did not decrease over time.

Pages 9-10, Lines 34-1:

The performance statistics of the CS19 gradient method for each cation are summarized in Table 1. The method shows high reproducibility, with method precisions better than 10 % for most analytes. Although the instrumental response varied from month-to-month for each analyte, this variability was random and the calibration curve slopes for each analyte showed no systematic decrease over time. The larger variability in the TMAH+ and TEAH+ calibration curves (+16 % and +11 % respectively) could likely be driven by their lower Henry's Law constants ($K_H$) in water (Christie and Crisp, 1967), resulting in volatilization losses from standards.

Comment 8: How long had the column been used when the degradation occurred? Is this method realistic to use for regular analysis if the column degrades with time. With so many peaks, often at low concentrations, it is critical to know when everything elutes.

Response 8: The column had been used nearly continuously for about a year before noticeable degradation occurred and more than 1000 samples and standards had been injected by the time the additional amines were added to the method. Provided that calibrations are performed regularly to ensure system performance is being tracked, the small amount of column degradation occurring over time is unlikely to present a major issue. In our experience, a one-minute shift of retention time typically occurs after so many injections, regardless of column stationary phase for either cations or anions when analyzing environmental samples, and the retention cannot be recovered by traditional clean-ups (VandenBoer et al., 2011). The following has been added to the manuscript to quantify the column degradation over time:

Page 12, Line 3-7:

The separation statistics for these additional amines are also presented in Table 1. Since the additional amines were injected after column degradation had occurred and retention times had noticeably shifted (see Fig. 3a vs. Fig. 1b and Sect 3.1.5 for further discussion), retention time and peak width were estimated using changes in separation parameters from the original method development for the methyl and ethyl amines.

Page 13, Lines 18-23:

Over the course of five months, peak retention times noticeably decreased and peak broadening of approximately 50 % occurred for all analytes. After more than 1000 sample and standard injections retention times had decreased by 1.9 + 0.1 minutes depending on the cation. Peak-to-peak resolution, however, remained largely unchanged throughout the column degradation during standard and sample analysis, even with observed peak broadening. This is consistent with what has been previously reported in the literature when hundreds to thousands of injections have been run through an IC column (VandenBoer et al., 2011).

Comment 9: Loss of Mg and Ca is likely related to the suppressor no longer working as it should. In my experience Mg and Ca are the first to go when a suppressor stops working. Was it replaced with a new one (that worked with a traditional CS12 method)? There could be changes to the sensitivity of other species if the suppressor stopped working properly, especially those with the later retention times. We've observed that if you aren't controlling the current properly for the low concentration of MSA in the eluent you can go through a lot of suppressors. The manual for the SRS300 provides guidance for calculating the correct current setting. Please discuss the suppressor setting and potential implications if it wasn't working as expected.

Response 9: Firstly, the authors would like to apologize for misrepresenting the issue with $Mg_{2+}$ and $Ca_{2+}$ in the discussion paper, as we have since determined that the method did not actually experience any loss of the divalent cations. Instead, $Mg_{2+}$ and $Ca_{2+}$ experienced severe peak broadening over time until they could not be resolved. After approximately 1000 injections, the combined unresolved peak had a peak width greater than 10 minutes. After integrating the data, the combined peak areas remained largely unchanged during the course of method development and all intermittent sample analysis. We conducted a full clean-up on the suppressor and replaced the eluent cartridge, concentrator column and CG19 guard column during troubleshooting to no avail, supporting the suppressor failure hypothesis. The suppressor current was set, at

all times, to values calculated using the equations provided in the CERS 500 operators manual, which are consistent with those generated in the Chromeleon method wizard. We agree with the Reviewer that in order to suppress our highest eluent concentration that we are over-suppressing the eluent for the majority of the run, which may decrease the suppressor lifetime. However, this is a necessity for separation of the alkyl amines from the inorganic cations.

With that said, the authors would like to thank the Reviewer for this suggestion, as replacing the suppressor was something we had not attempted in order to regain magnesium and calcium peak shape and resolution during months of troubleshooting. We could not find any definitive reports in the literature or indication from direct communications with the manufacturer that this was the cause of the loss of our peak shapes. Since submission of this manuscript, further issues with cation detection occurred and we have now replaced the CERS 500 and regained peak-to-peak resolution between Mg2+ and Ca2+. We have also observed a decrease in both analyte peak widths and a return of their original retention times. With the replacement of the suppressor we were able to finish calculating the method performance statistics for magnesium and calcium and have updated the manuscript with the following changes:

Page 13, Lines 26-35:

During the course of method development severe peak broadening and subsequent peak-to-peak resolution loss of magnesium and calcium was also observed. After the analysis of hundreds of samples the unresolved co-eluting divalent cations had a peak width greater than 10 minutes wide. The magnesium and calcium peak areas eventually became unresolved, with their combined peak area precision in the highest standard within 6 % (+ 1$\sigma$) after 12 months of analysis. It was determined that this broadening effect observed for the Mg2+ and Ca2+ peaks was due to a malfunctioning suppressor. After replacing the suppressor, a peak-to-peak resolution greater than one was restored for these analytes. Furthermore, the cumulative analyte peak broadening that had occurred throughout method development and sample analysis for all

the monovalent cations was also mitigated by installing the new suppressor. Retention times however were still shifted by 1.9 + 0.1 minutes, indicating that analytical column degradation had still occurred.

Page 12, Removed from manuscript:

Cleanup and eventual replacement of all IC components one by one was not able to resolve this issue. Since retention of magnesium and calcium on this column or others (e.g. CS12A) never returned, method statistics were calculated for the two cations using the limited calibrations and chromatograms prior to the occurrence of this system issue. Accuracy for both cations was unable to be assessed before peak loss occurred. Though loss of magnesium and calcium during the investigation was unfortunate and unexplained, it did not influence the separation of the analytes of interest, and the method was still successfully applied to real samples after the loss had occurred. Cleanup and eventual replacement of all IC components one by one was not able to resolve this issue. Since retention of magnesium and calcium on this column or others (e.g. CS12A) never returned, method statistics were calculated for the two cations using the limited calibrations and chromatograms prior to the occurrence of this system issue. Accuracy for both cations was unable to be assessed before peak loss occurred. Though loss of magnesium and calcium during the investigation was unfortunate and unexplained, it did not influence the separation of the analytes of interest, and the method was still successfully applied to real samples after the loss had occurred.

Comment 10: Please explain what the method blank is.

Response 10: We kindly refer the Reviewer back to the response to comment 5 where we addressed this issue.

Comment 11: Please include nominal concentrations of the cations and amines or someway to know if the change in peak height is a function of the method parameters or concentration. For example you suggest comparing Figure 3a and 1b (Pg 10 line

33) to see the shift in retention time - I can't help but notice the y-scales are very different. When you are losing peaks and seeing column degradation it is important to know if the peak height/area is also changing. The statements about the calibration curves suggest the areas didn't change over the course of your study but by including the concentration information for each chromatogram/injection you would clearly show that.

Response 11: The reviewer has raised a good point as the concentrations depicted in the figures are different and our reference is to retention time. To avoid confusion we have updated the captions for Figures 1 and 3 to include the injected analyte quantities as follows:

Pages 27 and 29:

Figure 1. Separation of amine and inorganic cation standards with the highest resolution gradient program at (a) 30°C and (b) 55°C. The order of elution and mass of cation injected in (a) is as follows: $Li+$ (1, 16 ng), $Na+$ (2, 158 ng), $NH4+$ (3, 169 ng), $MMAH+$ (4, 500 ng), $MEAH+$ (5, 500 ng), $K+$ (6, 524 ng), $DMAH+$ (7, 500 ng), $TMAH+$ (8, 500 ng), $DEAH+$ (9, 500 ng), $TEAH+$ (10, 500 ng), $Mg2+$ (11, 128 ng), and $Ca2+$ (12, 361 ng). Cation peaks represent the same mass injected and are labeled according to the same identities in (b).

Figure 3. (a) Separation of amine and inorganic cation standards with the addition of $MPAH+$, $iMPAH+$ and $MBAH+$ using the final gradient program. The order of elution and mass of cation injected in (a) is as follows: $Li+$ (1, 1.6 ng), $Na+$ (2, 16 ng), $NH4+$ (3, 17 ng), $MMAH+$ (4, 50 ng), $K+$ (5, 52 ng), $MEAH+$ (6, 50 ng), $DMAH+$ (7, 50 ng), $iMPAH+$ (8, 50 ng), $MPAH+$ (9, 50 ng), $TMAH+$ (10, 50 ng), $DEAH+$ (11, 50 ng), $MBAH+$ (12, 50 ng), and $TEAH+$ (13, 50 ng). (b) Separation of amine and inorganic cation standards with the addition of $MPAH+$, $iMPAH+$ and $MBAH+$ and the addition of the CG15 column using a modified gradient program. Cation peaks are labeled accordingly to the same identities in (b) and the mass of analyte injected is as follows: $Li+$ (1.6 ng), $Na+$ (16

ng), NH4+ (17 ng), MMAH+ (500 ng), K+ (52 ng), MEAH+ (500 ng), DMAH+ (500 ng), iMPAH+ (500 ng), MPAH+ (500 ng), TMAH+ (500 ng), DEAH+ (500 ng), MBAH+ (500 ng), and TEAH+ (500 ng).

Comment 12: In Figure 3b I like the separation of the types of amines. I think it would be helpful for the comparison between a and b to do the same for a, if possible. Or some type of color-coding by type.

Response 12: Figure 3a has been updated to include a similar color-coding scheme used in Figure 3b. We have also added a combined chromatogram that is offset from the individual components to demonstrate the overall method performance for the suite of analytes.

Comment 13: Figure 5a: add a summation sign in y-axis label to clarify it is the sum of the amines. And add in the caption it is the sum of the methyl and ethyl amines as you state in the text. Also add what the error bars represent. Figure 5 b: is this also ratio of the sum of the methyl and ethyl amines to ammonium?

Response 13: Both figures 5a and 5b have updated to include a summation sign in the y-axis and the caption for the figure has been updated as follows:

Page 31:

Figure 5. (a) Amines to ammonium ratio in the size-resolved aged biomass-burning sample originating from Quebec and Labrador in the summer of 2013. (b) Amines to ammonium ratio for the Burnaby/Kensington Park (BKP) site and North Vancouver/Second Narrows (NVSN) site in British Columbia during the summer 2015 wildfires. The error bars in the graph represent propagated error in the amine and ammonium quantities from field blank corrections and sample analysis check standard accuracies.

Comment 14: Table 2 seems more appropriate for the supplement.

Response 14: The authors agree and it has now been moved to the SI (Table S2).

Comment 15: Pg 4: 9-11 Please define all amines here so the definitions are easily found in one place.

Response 15: The following changes have been made to the manuscript:

Page 4, Lines 10-16:

Alkyl amines (MMA (monomethylamine, 40% w/w), DMA (dimethylamine, 40% w/w), TMA (trimethylamine, 25 % w/w), MEA (monoethylamine, 70 % w/w), DEA (diethylamine, >99% w/w), TEA (triethylamine, >99% w/w), MPA (monopropylamine, > 99% w/w), iMPA (isomonopropylamine, > 99% w/w) and MBA (monobutylamine, 99.5% w/w)).

Comment 15: Pg 7:2 Remove therefore

Response 15: We have made this change to the manuscript.

Comment 16: Pg 10: 15 Please define ppqv

Response 16: The following has been updated in the manuscript:

Page 11, Line 25:

Employing a method that is capable of quantifying amines at these very low mixing ratios is valuable since recent work has shown that parts per quadrillion by volume (ppqv) concentrations of gaseous amines can lead to particle growth (Almeida et al., 2013).

Comment 17: Pg 14: 1-2 Did the PM2.5 concentration need to be in excess of 200 ug/m3 for samples to be collected during the 8 hr period? Please clarify.

Response 17: During the collection of the BB samples in British Columbia the PM2.5 mixing ratio did not need to be in excess of 200 $\mu$g m-3, we were reporting the concurrent measurement of particle concentration during the sampling period. The sentence in the manuscript has been changed to reflect this:

[Figure]

Page 15, Line 37:

These PM2.5 samples were collected every 8 hours while the plume was traversing each site and during collection the PM2.5 concentration was measured independently to be in excess of 200 $\mu$g m-3.

Reviewer 2:

Comment 1: The authors point out, that the matrix of atmospheric samples can cause problems in the analytical measurements (which is true). Still I think that the problem of matrix effects is not too much discussed in the manuscript concerning this analytical method. For example on page 10, line 10/11 the authors write that using a pre-concentration column, you can eliminate negative charged and neutral disturbing ions. Did you test this here? And still the separation of the amines from other dominating cations that are for example present in marine samples in high quantities (sodium, ammonium) would be a problem – they would be enriched using a pre-column. This could affect again the separation of the (lower concentrated) amines.

Generally, I was wondering if you performed some further tests regarding matrix effects. Did you for example perform standard addition (to prove the accuracy of the method) with the field samples? This would be a high value in the method validation.

Response 1: We thank the Reviewer for the detailed question and are happy to elaborate here. A preconcentrating column acts identically to a cation-selective solid-phase extraction (SPE) column in that the stationary phase is selective for positively charged analytes while negative and neutral analytes are not retained. Such columns are commonly used in as a means of cleaning up sample matrices and thus we employ the use of an in-line concentrator column in our method to do the same. We did not test this anion and neutral exclusion property explicitly as these devices have been long in use as analytical sample cleanup tools (eg. Novič et al., 2001). The Reviewer is correct that with the use of a concentrating column all sample cations would be enriched, but not disproportionate to the original sample composition provided the capacity of the

column is not exceeded, which is the case for all data presented in this manuscript. To negate matrix and overloading effects, our samples were always diluted so the total quantity of cations was in the working calibration range of our method.

While we did not initially perform a standard addition, we agree that it is of high value in method validation and the ability to confirm the absence of matrix effects. Thus, we have extracted another sub-sample of biomass burning aerosols from our field collections in order to perform a standard addition and we have added a results table to the supporting information (Table S1). Since many of these same concerns were raised by Reviewer 1, all changes made to the manuscript that address these specific concerns are listed in the response to Comment 1 from Reviewer 1 above.

Comment 2: The authors use different types of field samples for the measurements of the amines with their method: nano Moudi aluminium foils and PM 2.5 filter samples. I was wondering if you considered measurements of field blanks, e.g. did the filter material and the sampling technique affect the sampling of the amines? Did you observe artifacts (e.g. from the gas phase) using the filter samples? This topic was addressed for example in Müller et al. (2009) in ACP that the authors cite here. Could you comment on this topic?

Response 2: Field blanks and method blanks were analyzed for both sets of BB samples but were not addressed in the submitted manuscript. We have described much of our approach in response to comment 5 from Reviewer 1 and elaborate further here for the Reviewer's additional concerns. There was no amine contamination present on any of the field blanks and all of the inorganic cation quantities were blank-corrected before being used to report atmospheric quantities of these analytes. The manuscript has been updated to reflect these statements.

With regards to the Reviewer's comments about sampling artifacts, the flow path of the atmospheric sample through the MOUDI minimizes gas-phase interactions with the sampling substrate that could lead to the accumulation of amines through sorption

processes. For the samples collected on the filter tape using the beta attenuation monitor, our selection of a field blank from the area outside the particle impaction zone would have been exposed to the gases in the sample flow, and thus our field blank corrections would have taken positive sorption biases into effect. This same positive sorption bias was noted by Müller et al. (2009) to which the Reviewer is referring. With respect to reactive incorporation of amines into the sample during the collection period, we do not have a method to quantify such a bias for the filter tape samples. Overall, the filter tape biomass burning samples presented here represents a time-integrated particulate composition reflecting the thermodynamic equilibrium between the gas and particulate phases over the duration of collection period of each sample.

The following additional changes have been made to the manuscript:

Page 14, Lines 7-14:

Particles released during forest fires have also been shown to contain highly oxidized large molecular weight organics (Di Lorenzo and Young, 2016; Saleh et al., 2014). We tested the robustness of our method on water-extracted aged biomass-burning particle samples collected by a cascade impactor in St. John's, Canada. Gas-sorption and reaction artifacts are minimized due to the gaseous flow-path being directed around the nanoMOUDI impaction plates, therefore samples analyzed are representative of the particles in the atmosphere. An overlaid chromatogram of two different size-resolved particle samples (100 - 180 nm and 320 – 560 nm) shows the presence of MMAH+, DMAH+ and DEAH+ in the aged biomass burning samples with complete separation from K+, NH4+ and Na+ (Fig. 4).

Page 16, Lines 7-20:

The detection of iMPAH+ has not previously been reported in biomass-burning emission inventories, and based on our measurements may be important to quantify in future controlled burn experiments. Our results differ from the study conducted by Takahama et al (2011) on the 2009 forest fires in British Columbia that reports the
detection of primary amine groups, which further suggest that amine emissions from biomass burning and/or their incorporation into biomass burning particles are not well understood. Although our observed suite of amines includes iMPAH+, there was no indication of other primary amines from the analyses of the BB particles. The calculated quantities for these samples may be biased high or low because the beta attenuation monitor used to sample the BB particles can be prone to gas-phase blow-on or blow-off artifacts via sorption or reactive mechanisms, as per traditional filter sampling strategies. Although the extracted field blank was corrected for any positive sorption biases on the filter tape (e.g. Müller et al., 2009), it was unable to account for any reactive uptake of gas-phase amines during the sampling period. Therefore, the filter tape BB samples presented here represent a time-integrated particulate composition assuming thermodynamic equilibrium between the gas and particulate phases for the duration of collection for each sample.

Comment 3: The authors report about the temperature effect in the IC separation. Also other authors have used temperatures above 50 âŮęC e.g. van Pinxteren et al., (J Atmos Chem (2015) 72: 105, doi:10.1007/s10874-015-9307-3), have used a column temperature of 60âŮęC in the IC separation of aliphatic amines. Maybe you can include previous publications that regard temperature effects in IC separation.

Response 3: The Reviewer is correct. We do not cite any papers that use high temperatures in their separation for field sample analysis and we have added in the van Pixteren et al manuscript (van Pinxteren et al., 2015). Many IC separations in the literature increase column temperature to change the separation efficiency and selectivity, which we have cited and discussed (Hatsis and Lucy, 2001; Rey and Pohl, 1996; Rey and Pohl, 2003). The CS19 analytical column is the only ThermoScientific IC column that is not meant to operate at temperatures as high as 60 °C. We have explained the established understanding of the effects of a higher temperature on cation selectivity to justify why this is explored in our ion chromatography optimization.

Page 8, Lines 29-37:

All gradient methods that were tested started with a 1 mM hold, followed by a step-wise increase and/or ramp to higher eluent concentrations at a column temperature of 30 °C. By combining the best isocratic separations for various pairs of cation analytes sequentially, iterative modifications were used to improve resolution based on Eq. (1-3) with the best separation method selected from amongst the iterations. Higher column temperature has been used to improve the quality of a separation method (Hatsis and Lucy, 2001; van Pinxteren et al., 2015; Rey and Pohl, 1996; Rey and Pohl, 2003). Therefore, column temperature was systematically increased to investigate if further improvement in peak-to-peak resolution was possible.

Comment 4: Page 8, line 32: ". . . have detected large quantities of MEA and DEA in ambient air. . ." do you refer to gas phase or aerosol measurements in the cited literature? Please specify.

Response 4: The following has been added to the manuscript:

Page 9, Line 20-22:

Multiple field campaigns have detected large quantities of gas-phase and particle-phase MEA and DEA in ambient air as well (Facchini et al., 2008; Müller et al., 2009; Sorooshian et al., 2009; Yang et al., 2005; Yang et al., 2004).

Comment 5: The authors discuss the problem of decreasing separation due to column instability. The authors mention that the high column temperature might accelerate this process. I am however sceptic if it is a good and practical solution to measure all samples at 30 ÅȩC and to repeat all measurements that indicate the presence of DEA or TEA at higher temperatures. Do you have any marker for column stability that indicate when you should replace the column? As the column degradation occurs gradually you might lose peak information in your samples quite soon (especially regarding field samples).

Response 5: During column degradation, which was experienced over the course of

our multiple analyses, retention times for the cations began to decrease. However, no peak area or peak height changes were observed over time with column degradation. There is no precise marker for column stability, but by routinely injecting calibration and check standards, column stability via peak separation efficiency can be monitored over time. Subjectively, the column used for our approach would be considered compromised when a resolution of 1.0 could no longer be maintained between our critical analyte peaks. We discuss this issue of column degradation in more detail in our response to comments 8 and 9 from Reviewer 1 above. All changes made to the manuscript to address this concern can also be found in our response to comments 8 and 9 from Reviewer 1.

Comment 6: Page 12: line 3 and following: The loss of calcium and magnesium. Could you speculate about the loss? Is it irreversible bound to the column surface? In the conclusion, you describe that ion chromatography methods often suffer from interferences with magnesium or calcium and the present work overcomes this problem. However, it cannot be the solution to have an "unfortunate and unexplained" loss of these elements. Did you consult the manufactures about this problem? What is your future strategy here?

Response 6: The authors agree with the Reviewer. We did not intend to leave this Mg and Ca discussion section as it is for the final manuscript, but rather were hoping that the Reviewers may suggest a potential solution or be aware of literature precedent to this issue for us to include in the final manuscript. We had certainly consulted the manufacturer on this issue, but they had no recommendations that worked (e.g. following the cleanup procedures in the manual, which we had already performed) and they did not have diagnostics that could definitively isolate our suppressor as the source of this issue. We hope that by publishing this finding in our revised manuscript, that this issue can be more easily identified and resolved in other labs that encounter such ion losses. This issue has now been addressed in detail in response to comment 9 from Reviewer 1. In short, after careful examination of the chromatograms, we noticed that

Mg2+ and Ca2+ were not in fact lost during the period over which this method was developed. Instead, the two divalent cations increasingly broadened until they eluted as one broad unresolved peak. This peak widened over time and after a thousand injections was more than 10 minutes wide. This demonstrates that Mg and Ca were not being irreversibly bound to the column surface or any other IC component during method development and instrumental performance assessment. Upon replacement of the suppressor, as per Reviewer 1's suggestion, we were able to regain resolution of magnesium and calcium, again demonstrating that they were not being irreversibly bound to the column. The manuscript has also now been updated to reflect these new findings and also elaborate on the Mg and Ca method shortcomings in the response to comment 9 from reviewer 1.

Comment 7: Page 12, line 37: unclear phase; did you mean ". . . in the size range of 100 to 560 nm"? Please clarify.

Response 7: Yes, this is what we meant. The manuscript has been clarified.

Comment 8: Page 13, line 21 and following: The high ratio of amines to ammonium is very surprising. As one explanation, the authors state that marine influences could be an explanation for this observation. While amines can indeed have significant marine sources (as also stated by the authors), also ammonium would be elevated then (as also shown in the cited literature, e.g. Gibb et al). So do you think that there could be a marine source producing so much more amines than ammonium? Is there any evidence in the literature?

Response 8: Unfortunately, it is outside the scope of the discussion of this new methodology and its performance presented here. We are currently exploring the nature of more comprehensive chemical characterization with a broader context to understand what the amines to ammonium ratio in these samples may mean. This will be discussed further in a separate publication focused on thermodynamic partitioning expectations for aerosol composition and the potential reactivity of organic matter released from

biomass burning plumes with reduced nitrogen compounds, such as amines. For now, we do not suggest that there is a marine source producing a larger quantity of amines than ammonium, but would be willing to state that there appears to be preferential uptake of amines to the aerosol in the biomass burning plume as all amines sources tend to coincide with emissions of ammonia in greater quantity. The potential marine source is supported by the composition of the samples collected in BC where the BB plume did not interact with coastal marine air masses. We have added the following to the manuscript to further clarify our position on the marine amines hypothesis:

Page 14, Lines 26-33:

The high concentrations of DMA and DEA produced by marine biological activity could then partition into the biomass burning particles and react to neutralize inorganic acids (e.g. sulfuric acid), form salts or amides with organic acids, or react with carbonyl moieties in the highly oxidized organic material produced via BB to form imines (Qiu and Zhang, 2013). If this explanation holds true then there may be preferential uptake of these amines over ammonia into the plume, as there is no evidence yet that suggests a larger agricultural or marine source of amines relative to ammonia to the atmosphere. This marine amines hypothesis, while consistent with observations in the literature, is beyond the scope of this work in terms of assigning the DMA and DEA source.

Comment 9: The authors might consider including the recent publication of van Pinxteren et al. 2015, as they reported high concentration of the amines in a marine region close to the Cape Verdean islands.

Response 9: Yes, we did not come across this publication during our literature review and agree that it is valuable, and has been added to add to the manuscript.

Comment 10: Did you measure other marine tracers (like MSA) to test the hypothesis?

Response 10: No we did not, however given that MSA is commonly used as a marine tracer in the literature it is something we may consider to try and quantify in the future

with our anion IC methods. We did quantify the amount of sea-salt sulphate present in the particle samples and again, as this is outside the scope of a methodological manuscript, are planning to discuss its relationship to the alkyl amines in a forthcoming manuscript.

———————————————

[Figure]

**Supplement:**

[revised manuscript text omitted]
 targeting these reduced nitrogen species. Being able to chromatographically resolve alkyl amines from the dominant base, ammonium, represents a major challenge when sampling the gas phase (Chang et al., 2003; Ge et al., 2011; Schade and Crutzen, 1995). Quantifying amines in particle samples, for example by ion chromatography, presents a greater challenge due to possible interferences from sodium, potassium, ammonium, magnesium and calcium whose concentrations are dependent on the particle source characteristics and the measurement location (Ault et al., 2013; Kovac et al., 2013; Sobanska et al., 2012; Sun et al., 2006). Particles frequently contain complex organic mixtures, such as high molecular weight organic compounds, which can cause further matrix effects during separation or direct analysis (Di Lorenzo and Young, 2016; Saleh et al., 2014).

[revised manuscript text omitted]

Optimal separation of a suite of 15 cations was achieved using a mobile phase flow rate of 1.25 ml min$^{-1}$ and a column temperature of 55 °C. The eluent gradient program is as follows: an initial MSA concentration of 1 mM held for 20 minutes, a step increase to 4 mM followed immediately by an exponential ramp to 10 mM over 10 minutes (Chromeleon curve factor = 7). The final concentration of 10 mM was held for an additional 5 minutes, yielding a total run time of 35 minutes. The IC was returned to initial conditions and re-equilibrated for 10 minutes as the next 1 ml sample aliquot was prepared for injection by the AS-DV. The suppressor current, optimized for this flow rate and the maximum eluent concentration in accordance with the calculation provided by the manufacturer, was set at 37 mA. The typical backpressure in the system at these conditions was 2100 psi. The Chromeleon method file for the method described above is detailed in Fig. S1.

**2.4    Quality Assurance and Quality Control**

Standards were prepared using Class A Corning polymethylpentene 50 ($\pm$ 0.06) ml volumetric flasks that were rinsed 4 times with ethanol and 8 times with ultrapure water prior to use. Standards were stored in 60 ml brown Nalgene polypropylene bottles that were pre-cleaned in a 10 % HCl bath, followed by 8 sequential rinses with distilled and ultrapure water, respectively. The mixed amine standards and mixed inorganic cation standards were prepared separately and each cation standard set was composed of five calibration standards, two check standards and an ultrapure deionized water blank. Ranges and related parameters are denoted by mass injected, as the preconcentration column negates the effect of volume. All amine calibration standards had a mass calibration range of 5-500 ng. The mass range for each inorganic cation calibration is as follows: Li$^+$ (0.82-16 ng), Na$^+$ (7.8-160 ng), NH$_4^+$ (8.4-170 ng), K$^+$ (26-520 ng), Mg$^{2+}$ (6.4-130 ng), and Ca$^{2+}$ (18-360 ng). All calibration curves contained trace inorganic cation impurities from the ultrapure deionized water source or holding vessels that fell below the lowest calibration standard, and were corrected accordingly to allow for inter-day method performance comparison. Trace quantities of amines were not observed. An example of a calibration blank chromatogram is presented in Fig. S2.

Method precision for each methyl and ethyl amine cation was determined using standard calibration curves (n = 9) injected across five different days spanning three months. The precision for the propyl and butyl amines was determined using two standard calibration curves analyzed over one month. Method precision for Li$^+$, Na$^+$, NH$_4^+$ and K$^+$ was assessed using calibrations (n = 6) from three separate days spanning two months. Precision for each cation was calculated using the standard deviation ($\sigma$) in the slope of the linear calibration curves. Check standards positioned between the two highest and the two lowest calibration standards for each cation were used to determine method accuracy across the calibration range. The low check standard was 15 times greater than the lowest standard and the high check standard was 150 times higher than the lowest standard. Accuracy was determined by

the percent relative error between the known and calculated concentrations of the check standards. The limits of detection (LOD) for the singly-charged inorganic cations (n = 4) and methyl and ethyl amines (n = 5) were determined using calibration standard and calibration blank chromatograms from three or more separate days. The LODs for the propyl and butyl amines were determined using calibration standard and blank chromatograms from two separate days. The LODs are reported as concentrations resulting in a signal peak height to background noise ratio of three. The background noise was determined using the standard deviation of the conductance signal that fell within the retention time window for each analyte in their respective calibration blank chromatograms.

To assess the method robustness in the presence of a complex matrix the gradient method standard addition was performed on a subsample of a size-resolved BB particle extract (320 – 560 nm) (see Sect 2.5). Standard addition was performed by adding known quantities of methyl and ethyl amine solution to a 0.5 ml sample of the extract followed by dilution to 5 ml. The amount of the methyl and ethyl amines added to the internal calibration matched that of the external calibration. The slope and retention times for the methyl and ethyl amines from the internal calibration were calculated and compared to those performed externally to quantify matrix effects. Discussion of the analytical performance of the CS19 gradient program is presented in Sect. 3.1.2.

**2.5    Size-resolved BB sample analysis**

A size-resolved particle sample from a BB plume was collected using a nanoMOUDI II (nano micro-orifice uniform-deposit impactor, model 122-R, MSP Corp., Shoreview, MN, USA) in St. John's, Newfoundland on July 6, 2013. Satellite images of the plume smoke, HYSPLIT back trajectories, as well as, measured $PM_{2.5}$ concentrations reported by Environment and Climate Change Canada indicate that these plumes originated from boreal forest fires in northern Quebec and Labrador on July 4, 2013 and travelled via Labrador and the Gulf of St. Lawrence to the sampling site (Di Lorenzo and Young, 2016). The nanoMOUDI samples were collected on 13 aluminum substrate stages into size-resolved bins of atmospheric particles with a diameter range spanning 0.010 –18 μm. Air was sampled continuously for 25.5 hours at a flow rate of 30 L min$^{-1}$. A sub-sample of each aluminum substrate (10 % of the total substrate area) was extracted into a glass vial with 5 mL ultrapure deionized water by sonication (VWR Scientific Products/Aquasonic 150HT, Ultrasonic Water Bath) for 40 minutes. The extracts were filtered using a 0.2 μm polytetrafluoroethylene (PTFE) filter and stored in polypropylene vials at 4 °C prior to analysis by IC within 24 hours. Cation analytes within these samples all fell within their respective calibration ranges and did not require any further dilution. An aluminum substrate field blank was also transported and exposed to the ambient atmosphere briefly at the collection site before being stored in a sealed container for the duration of the sample collection, transported back with the samples, and extracted simultaneously following the same procedure. All calculated quantities were corrected with measurements of the field blank with the additional error from this correction propagated into our final reported values. The field blank chromatogram for the size-resolved BB samples is presented in Fig. S2.

**2.6    BC fire sample analysis**

[revised manuscript text omitted]

20     rate to quantify values of peak-to-peak resolution for the inorganic cations and alkyl amines before approaching a

21     gradient method (Fig. S3 and S4). An increase in resolution greater than one was observed for all analyte pairs

22     aside from DEAH$^{+}$/TMAH$^{+}$ when using a 1 mM MSA eluent concentration.

23     All gradient methods that were tested started with a 1 mM hold, followed by a step-wise increase and/or ramp to

24     higher eluent concentrations at a column temperature of 30 °C. By combining the best isocratic separations for

25     various pairs of cation analytes sequentially, iterative modifications were used to improve resolution based on Eq.

26     (1-3) with the best separation method selected from amongst the iterations. Higher column temperature has been

27     used to improve the quality of a separation method (Hatsis and Lucy, 2001; van Pinxteren et al., 2015; Rey and

28     Pohl, 1996; Rey and Pohl, 2003). Therefore, column temperature was systematically increased to investigate if

29     further improvement in peak-to-peak resolution was possible. Temperature effects on separation efficiency in ion

30     chromatography are thermodynamically complex (Hatsis and Lucy, 2001; Kulis, K., 2004; Rey and Pohl, 1996),

31     but typically result in increased peak resolution because of improvements in mobile phase diffusivity, which

32     increases the efficiency term from Eq. (2). Higher temperatures can replicate the separation effects observed when

33     adding an organic mobile phase modifier (Hatsis and Lucy, 2001; Rey and Pohl, 1996). Figures 1a and b show

34     gradient separations at 30 °C and 55 °C respectively. At 30 °C, K$^{+}$ and DMAH$^{+}$ overlap considerably (R$_S$ = 0.45)

35     and DEAH$^{+}$ and TMAH$^{+}$ coelute. By increasing the column temperature to 55 °C, the extent of peak overlap

36     between K$^{+}$ and neighboring alkyl amine cations is noticeably reduced (R$_S$ > 1) and DEAH$^{+}$ and TMAH$^{+}$ are

37     increasingly well resolved (R$_S$ = 1.48). The effect of temperature on the separation of the alkyl amines is

demonstrated in Fig. 2, where the separation of DEAH$^+$ with TMAH$^+$ is achieved above 50 °C. The temperature increase also results in lower resolution between DMAH$^+$ and MEAH$^+$ from $R_S$ = 1.57 to $R_S$ = 1.08. In our method, a column temperature of 55 °C produced peak-to-peak resolutions greater than a value of one between all six alkyl amine cations and inorganic cations in the final gradient method, giving a 95 % separation between our target analytes and expected atmospheric interferences in the condensed phase. The peak-to-peak resolutions are summarized in Table 1. These represent a lower-limit in peak resolution since they were calculated using peak parameters at the upper limit of the working range, which was determined based on maximum mixing ratios or mass loadings expected for the analysis of atmospheric samples containing these analytes.

The separation method produced in this work is able to overcome previously reported IC coelution difficulties between DEAH$^+$ and TMAH$^+$ and between MEAH$^+$ and DMAH$^+$ (VandenBoer et al., 2012; Verriele et al., 2011). Both DMA and TMA have been identified as dominant amines in emission studies, so it is important to achieve accurate and specific quantitation of both species in gas and particulate atmospheric samples (Facchini et al., 2008; Kuwata et al., 1983; Müller et al., 2009; Van Neste and Duce, 1987). Multiple field campaigns have detected large quantities of gas-phase and particle-phase MEA and DEA in ambient air as well (Facchini et al., 2008; Müller et al., 2009; Sorooshian et al., 2009; Yang et al., 2005; Yang et al., 2004). In some cases clean up steps have been used to alleviate IC interferences from common atmospheric cation species in the quantitation of amines despite the fact that an 85% evaporation loss of the amines, in addition to the extra sample handling, was reported when using a solid phase extraction clean up (Huang et al., 2014). The CS19 IC method reported here is able to separate the most common atmospheric inorganic cations in addition to eleven common atmospheric amines (see additional separation of five additional alkyl amines in Sect 3.1.3). It can be easily applied to water-soluble atmospheric gas and particulate samples since they can be directly analyzed - without coelution or a clean-up step - with separation times of similar duration to many previously reported methods, including those employing an online IC method (Huang et al., 2014; Murphy et al., 2007; VandenBoer et al., 2012; Verriele et al., 2011).

**3.1.2 Instrumental performance and comparison for the methyl amines, ethyl amines and inorganic cations**

The performance statistics of the CS19 gradient method for each cation are summarized in Table 1. The method shows high reproducibility, with method precisions better than 10 % for most analytes. Although the instrumental response varied from month-to-month for each analyte, this variability was random and the calibration curve slopes for each analyte showed no systematic decrease over time. The larger variability in the TMAH$^+$ and TEAH$^+$ calibration curves ($\pm$ 16 % and $\pm$ 11 % respectively) is likely driven by their lower Henry's Law constants ($K_H$) in water (Christie and Crisp, 1967), resulting in volatilization losses from standards. Concurrently, this variability could be driven by partitioning losses along the flow path, particularly when the tri-subsituted amines reach the suppressor, which was not temperature-controlled. In future investigations it may be worthwhile to acidify the standards to ensure the amines are maintained in their charged form in the aqueous phase. Alternatively, to combat losses to neutral forms, use of a Salt Converter suppressor accessory (ThermoScientific, SC-CSRS 300, P/N: 067530), which keeps weak electrolytes in a separated sample fully protonated prior to their conductance measurement, may also aid in increasing long-term TMAH$^+$ and TEAH$^+$ precision.

The limits of detection (LODs) for each analyte are reported in Table 1 as both a range and as the average LOD ($\pm$ 1$\sigma$). The LODs are reported in this manner to reflect the high day-to-day variability in the calculated LODs. This variability may be driven by i) the purity of the deionized water used for eluent generation; ii) instrumental baseline noise and trace contamination on the day of analysis; and iii) quality of labware cleaning prior to preparation of calibration blanks. Outliers in the LOD dataset were found to result from trace contamination of analytical labware, sampling vials, or from systematic errors made in the preparation of standards or injection of samples on the IC (e.g. leaking autosampler caps, failing retention of concentrator column). The Grubb's test was performed using a 95% confidence interval to statistically identify outliers from LOD data sets. Calculated detection limits were determined to lie in the picogram per injection range for all analytes. The LODs for the inorganic cations were 10 to 100 times lower than those of the alkyl amines, with the exception of $Mg^{2+}$ and $Ca^{2+}$. Trace contamination of calcium in our ultrapure deionized water led to higher LODs for the divalent cations. Our method shows high accuracy in the upper range of the calibrations for the methyl and ethyl amines, with accuracies ranging from 90 – 100 %. The accuracy was much lower for each methyl and ethyl amine cation at the low end of the calibration range where amine concentrations were approximately 1.5 times the limit of quantitation (LOQ). Quantitation near the method LOQ was more sensitive to small integration changes, which affected the calculated peak area, even when performing integrations manually, and this resulted in greater method error. This is a drawback inherent to IC since wide analyte peaks are a result of persistently large stationary phase particle sizes, causing band broadening via longer flow paths and increased diffusion during separation (i.e. the A and B terms in Eq. (3)). The low alkyl amine accuracies may also be driven by their air-water partitioning properties, which could result in losses during sample handling and during sample injection. The low and high range accuracies for all inorganic cations, with the exception of ammonium, were high (80 – 120%) because concentrations were not near the limit of quantitation for these analytes. The low check standard accuracy for ammonium likely arises due to similar issues as those discussed above for the tri-substituted amines.

To further test the efficacy of the separation method, a standard addition calibration was performed in the presence of the complex BB matrix. The calibration slopes and retention times for each analyte from the standard addition and external calibration performed on the same day are listed in Table S1. The slopes for the two calibrations varied between 0 and 8 %, which is within the method calibration precisions presented in Table 1. Thus, the BB sample extracts did not exhibit matrix effects as the slopes were found to be within the expected instrument variability as determined by external calibration standard analysis. However, increasing retention times of approximately 0.3 – 0.5 minutes were observed for all cation analytes when performing the standard addition. This is an effect inherent in IC when samples with higher total quantities of cations are preconcentrated, resulting in a sample plug filling a greater volume of the stationary phase capacity. The initial weak mobile phase of the gradient method will therefore take a greater amount of time to elute all of the analyte cations from the preconcentration and analytical columns. This same increase in retention times is present in the external calibration with increasingly concentrated standards (Table 1).

Previous IC instrumental precisions reported for use in quantifying the six atmospheric methyl and ethyl amines range from 0.4 to 17.2 %, which are comparable to our method (Table S2; Chang et al., 2003; Dawson et al., 2014; Erupe et al., 2010; Huang et al., 2014; Li et al., 2009; VandenBoer et al., 2012; Verriele et al., 2012;). Our separation method shows greater average variability than others due to our numerous assessments (n = 9) over

multiple months, a more comprehensive analysis compared to previous reports. The sensitivity of this instrument is also similar to that of all other reported IC methods as the instrumental detection limits are in the picogram range. Only VandenBoer et al. (2012) and Chang et al. (2003) report lower detection limits and these are likely a result of a lower background signal from running the IC instruments online. Our method does not achieve instrumental limits of detection as low as those achieved using derivatization methods coupled with GC-MS or HPLC analysis (Akyüz, 2007; Fournier et al., 2008; Possanzini and Di Palo, 1990). However, multi-step derivatization methods are prone to losses that must be quantified with internal standards. These losses can lead to higher overall method detection limits, which is not the case for direct analysis of water-soluble samples. Derivatization methods are also difficult to employ for near-real-time analyses of the atmosphere, making the approach less analytically attractive. Further, the IC separation method presented here is able to address additional matrix effects that may result from other atmospheric species through the use of a sample pre-concentration column. Only positively charged species are retained in this pre-concentration step and injected through the IC system for analysis, negating matrix effects from non-charged and anion species, as demonstrated by the standard addition to the BB sample extract.

Employing a method that is capable of quantifying amines at very low mixing ratios is valuable since recent work has shown that parts per quadrillion by volume (ppqv) concentrations of gaseous amines can lead to particle formation and growth (Almeida et al., 2013). If our method were applied to online atmospheric ambient sampling of gases or particles the method could be used to detect amines at ppqv mixing ratios. For example, a detectable signal for 100 ppqv mixing ratios could be attained by sampling through a bubbler, filter, or denuder at a low flow rate of 3 L min$^{-1}$ for 1 – 10 hours, depending on the amine. It may be possible to shorten the sample collection duration to an hourly timescale to detect ppqv mixing ratios of atmospheric amines if the method is interfaced with a high sensitivity MS detector. Verriele et al. (2014) observed a 5 – 30 fold improvement in method detection limits when interfacing their IC method with a MS detector.

**3.1.3  Expanded amine catalogue for other common atmospheric species**

[revised manuscript text omitted]

**3.1.5    Analytical column stability**

Over the course of five months, peak retention times noticeably decreased and peak broadening of approximately 50 % occurred for all analytes. After more than 1000 sample and standard injections retention times had decreased by 1.9 $\pm$ 0.1 minutes depending on the cation. Peak-to-peak resolution, however, remained largely unchanged throughout the column degradation during standard and sample analysis, even with observed peak broadening. This is consistent with what has been previously reported in the literature when hundreds to thousands of injections have been run through an IC column (VandenBoer et al., 2011). This may also be a result of column degradation from operating the CS19 column at a temperature higher than that recommended by the manufacturers.

During the course of method development severe peak broadening and subsequent peak-to-peak resolution loss of magnesium and calcium was also observed. After the analysis of hundreds of samples the unresolved co-eluting divalent cations had a peak width greater than 10 minutes wide. The magnesium and calcium peak areas eventually became unresolved, with their combined peak area precision in the highest standard within 6 % ($\pm$ 1σ) after 12 months of analysis. It was determined that this broadening effect observed for the $Mg^{2+}$ and $Ca^{2+}$ peaks was due to a malfunctioning suppressor. After replacing the suppressor, a peak-to-peak resolution greater than one was restored for these analytes. Furthermore, the cumulative analyte peak broadening that had occurred throughout method development and sample analysis for all the monovalent cations was also mitigated by installing the new suppressor. Retention times however were still shifted by 1.9 $\pm$ 0.1 minutes, indicating that analytical column degradation had still occurred.

**3.2    Biomass-burning particle analysis and discussion**

[revised manuscript text omitted]

6 30°C and (b) 55°C. The order of elution and mass of cation injected in (a) is as follows: $Li^+$ (1, 16 ng), $Na^+$ (2, 158

7 ng), $NH_4^+$ (3, 169 ng), $MMAH^+$ (4, 500 ng), $MEAH^+$ (5, 500 ng), $K^+$ (6, 524 ng), $DMAH^+$ (7, 500 ng), $TMAH^+$

8 (8, 500 ng), $DEAH^+$ (9, 500 ng), $TEAH^+$ (10, 500 ng), $Mg^{2+}$ (11, 128 ng), and $Ca^{2+}$ (12, 361 ng). Cation peaks

9 represent the same mass injected and are labeled according to the same numeric identities in (b).

[Figure]

Figure 2. Separation of 1 μg ml$^{-1}$ mixed amines standard with the final method gradient elution program at 30 °C, 40 °C, 50 °C and 60 °C. The peak elution order was MMAH$^{+}$ (1), MEAH$^{+}$ (2), DMAH$^{+}$ (3), TMAH$^{+}$ (4), DEAH$^{+}$ (5), and TEAH$^{+}$ (6). The separation of diethylamine (DEA) from trimethylamine (TEA) was achieved at column temperatures greater than 50 °C.

[Figure]

2
3
4    Figure 3. (a) Separation of amine and inorganic cation standards with the addition of MPAH$^+$, iMPAH$^+$ and
5    MBAH$^+$ using the final gradient program. The order of elution and mass of cation injected in (a) is as follows: Li$^+$
6    (1, 1.6 ng), Na$^+$ (2, 16 ng), NH$_4^+$ (3, 17 ng), MMAH$^+$ (4, 50 ng), K$^+$ (5, 52 ng), MEAH$^+$ (6, 50 ng), DMAH$^+$ (7, 50
7    ng), iMPAH$^+$ (8, 50 ng), MPAH$^+$ (9, 50 ng), TMAH$^+$ (10, 50 ng), DEAH$^+$ (11, 50 ng), MBAH$^+$ (12, 50 ng), and
8    TEAH$^+$ (13, 50 ng). (b) Separation of amine and inorganic cation standards with the addition of MPAH$^+$, iMPAH$^+$
9    and MBAH$^+$ and the addition of the CG15 column using a modified gradient program. Cation peaks are labeled
10   accordingly to the same identities in (b) and the mass of analyte injected is as follows: Li$^+$ (1.6 ng), Na$^+$ (16 ng),
11   NH$_4^+$ (17 ng), MMAH$^+$ (500 ng), K$^+$ (52 ng), MEAH$^+$ (500 ng), DMAH$^+$ (500 ng), iMPAH$^+$ (500 ng), MPAH$^+$
12   (500 ng), TMAH$^+$ (500 ng), DEAH$^+$ (500 ng), MBAH$^+$ (500 ng), and TEAH$^+$ (500 ng).

[Figure]

Figure 4. Overlaid chromatograms of MOUDI size-fractionated particle samples collected in St John's on July 6, 2013 during the intrusion of a biomass-burning plume that originated from Northern Labrador and Quebec. The robustness of the separation method for MMAH$^+$, DMAH$^+$ and DEAH$^+$ from the common inorganic cations is demonstrated for the 320-560 nm (Black) and 100-180 nm (Red) size bins.

[Figure]

[Figure]

Figure 5. (a) Amines to ammonium ratio in the size-resolved aged biomass-burning sample originating from Quebec and Labrador in the summer of 2013. (b) Amines to ammonium ratio for the Burnaby/Kensington Park (BKP) site and North Vancouver/Second Narrows (NVSN) site in British Columbia during the summer 2015 wildfires. The error bars in the graph represent propagated error in the amine and ammonium quantities resulting from variability in the field blanks and check standards during the analysis of the samples.

Table 1. Separation characteristics and statistics for the CS19 gradient method. The retention time ($t_r$) ranges for the methyl amines, ethyl amines and inorganic cations were determined using retention time windows from a full calibration. The peak width and resolution were determined using the highest calibration standards amines (500 ng) and inorganic (160 – 520 ng) cations. The $t_r$ range and peak width were back-calculated for iMPAH+, MPAH+, MBAH+, DABH+ and DAPH+ based on the other alkyl amines responses to column degradation. Sensitivity, precision, average LOD, and LOD range were analyzed using multiple calibration standards and blanks (see Section 2.4). Upper and lower range accuracies were assessed using high and low check standards for the alkyl amines (n = 6) and inorganic cations (n = 4). The low check standards were 15 times more concentrated than the lowest calibration standard and the high check standards were 150 times more concentrated.

| Cation | $t_r$ (min) | Peak width (min) | Resolution | Sensitivity ($\mu S*min*mol^{-1}$) | Precision % ($\pm 1\sigma$) | Upper range accuracy (%) | Lower range accuracy (%) | Average LOD (pg) ($\pm 1\sigma$) | LOD range (pg) |
|---|---|---|---|---|---|---|---|---|---|
| Li+ | 8.2 – 8.4 | 0.72 | 2.68 | 11.5E08 | 2 | 96 ± 5 | 82 ± 4 | 0.6 ± 0.2 | 0.3 - 0.8 |
| Na+ | 10.1 -10.3 | 0.73 | 1.87 | 5.04E08 | 2 | 95 ± 4 | 90 ± 6 | 8 ± 4 | 4 - 14 |
| NH4+ | 11.8 – 12.1 | 1.18 | 0.65/1.85* | 2.45E08 | 4 | 103 ± 4 | 50 ± 50 | 22 ± 17 | 7 - 47 |
| MEtAH+ | 12.7 – 13.0 | 0.99 | 0.56 | 2.2E08 | --- | --- | --- | 3600 (n = 1) | --- |
| MMAH+ | 13.5 – 13. 8 | 0.86 | 1.09 | 1.42E08 | 5 | 98 ± 6 | 40 ± 30 | 300 ± 300 | 30 - 650 |
| K+ | 14.7 – 14.8 | 0.87 | 1.22 | 4.14E08 | 5 | 99 ± 2 | 94 ± 7 | 14 ± 11 | 4 - 28 |
| MEAH+ | 15.5 – 15.8 | 0.79 | 1.08 | 0.90E08 | 7 | 97 ± 5 | 40 ± 10 | 500 ± 200 | 200 - 700 |
| DMAH+ | 16.4 – 16.7 | 0.85 | 1.64 | 1.48E08 | 5 | 100 ± 10 | 30 ± 30 | 200 ± 300 | 40 - 650 |
| iMPAH+ | 18.2 – 18.5 | 0.79 | 2.24 | 0.84E08 | 4 | 90 ± 10 | 80 ± 80 | 70 ± 40 | 40 - 90 |
| MPAH+ | 20.1 – 20.4 | 0.88 | 1.55 | 0.62E08 | 12 | 88 ± 4 | 90 ± 90 | 50 ± 40 | 20 - 80 |
| TMAH+ | 21.2 -21.6 | 1.11 | 1.48 | 0.34E08 | 16 | 90 ± 10 | 30 ± 20 | 600 ± 300 | 300 - 1200 |
| DEAH+ | 22.6 – 22.7 | 1.18 | 2.51 | 0.76E08 | 8 | 97 ± 8 | 50 ± 30 | 400 ± 300 | 100 - 800 |
| MBAH+ | 25.3 – 25.6 | 0.43 | 3.12 | 0.62E08 | 1 | 80 ± 20 | 100 ± 80 | 910 ± 30 | 890 - 930 |
| TEAH+ | 27.3 – 27.7 | 0.95 | 3.40 | 0.85E08 | 12 | 96 ± 4 | 49 ± 6 | 800 ± 400 | 500 - 1400 |
| Mg2+ | 30.4 – 30.8 | 0.79 | 1.22 | 12.2E08 | 1 | 80 ± 20 | 100 ± 30 | 2000 ± 3000 | 200 - 4000 |
| Ca2+ | 31.6 – 32.9 | 1.05 | 3.16 | 14.3E08 | 2 | 90 ± 20 | 120 ± 20 | 3700 ± 200 | 3500 – 3800 |
| DABH+ | 36.6 – 36.9 | 1.48 | 0.98 | 4.5E08 | --- | --- | --- | 1000 (n = 1) | --- |
| DAPH+ | 38.0 – 38.4 | 1.60 | N/A | 4.9E08 | --- | --- | --- | 180 (n = 1) | --- |

* = resolution calculated for NH4+ and MMAH+

1      --- = insufficient replicates

---

## Author Comment (AC2) · 1 Feb 2017

```
{Initial Time}    Instrument Setup
        Pump_ECD.Pressure.UpperLimit              3000 [psi]
        Sampler.DelayVolume                       125 [µl]
        Pump_ECD.%A.Equate                        "MSA"
        Pump_ECD.Pressure.LowerLimit              200 [psi]
        Pump_ECD.ColumnTemperature.Nominal        55.0 [°C]
        Pump_ECD.CellTemperature.Nominal          30.0 [°C]
        Pump_ECD.Data_Collection_Rate             5.0 [Hz]
        Pump_ECD.Suppressor_Type                  CERS_4mm
        Pump_ECD.Suppressor_MSA                   10.0 [mM]
        Pump_ECD.Suppressor_RecommendedCurrent    37 [mA]
        Pump_ECD.Suppressor_Current               37 [mA]
        Pump_ECD.CR_TC                            On
        Sampler.FlushFactor                       1
        Sampler.DeliverSpeed                      1.0 [ml/min]
        Sampler.InjectPosition
        Sampler.DeliverRinse                      500,Position=Rinse
        Sampler.DeliverSample                     Volume=Bleed
        Sampler.LoadPosition
        Sampler.DeliverSample
        Sampler.EndSamplePrep
        Pump_ECD.Flow                             1.25
        Pump_ECD.EluentGenerator_2.Concentration  1.00 [mM]
-5.000  Equilibration                             Duration = 5.000 [min]
        Pump_ECD.EluentGenerator_2.Concentration  1.00 [mM]
0.000   Inject
        Wait    Sampler.CycleTimeState,
                Run=Hold,
                Timeout=Infinite
        Sampler.Inject
        Start Run
        Pump_ECD.Channel_Pressure.AcqOn
        Pump_ECD.Autozero
        Pump_ECD.ECD_1.AcqOn
        Pump_ECD.ECD_Total.AcqOn
        Run                                       Duration = 45.000 [min]
        Pump_ECD.EluentGenerator_2.Concentration  1.00 [mM]
        Pump_ECD.EluentGenerator_2.Curve          5
20.000
        Pump_ECD.EluentGenerator_2.Concentration  1.00 [mM]
        Pump_ECD.EluentGenerator_2.Curve          7
        Pump_ECD.EluentGenerator_2.Concentration  4.00 [mM]
        Pump_ECD.EluentGenerator_2.Curve          7
30.000
        Pump_ECD.EluentGenerator_2.Concentration  10.00 [mM]
        Pump_ECD.EluentGenerator_2.Curve          7
40.000
        Pump_ECD.EluentGenerator_2.Concentration  10.00 [mM]
        Pump_ECD.EluentGenerator_2.Curve          5
        Pump_ECD.EluentGenerator_2.Concentration  1.00 [mM]
        Pump_ECD.EluentGenerator_2.Curve          5
45.000  Stop Run
        Pump_ECD.Channel_Pressure.AcqOff
        Pump_ECD.ECD_1.AcqOff
        Pump_ECD.ECD_Total.AcqOff
End
```

Fig S1. Chromeleon method data file for final gradient method at 55 °C

[Figure]

Fig S2. Sample chromatograms of a calibration blank and a size-resolved BB MOUDI foil substrate field blank. The peaks labelled above are as follows: $Na^+$ (1), $K^+$ (2), System peak (3), and $Ca^{2+}$ (4).

10

[Figure]

5    Fig S3. (a) Resolution of the six inorganic cation peak pairs using isocratic eluent methods at a flow rate of 0.75 ml min⁻¹. (b) Resolution of the six inorganic cation peak pairs using isocratic eluent methods at a flow rate of 1.25 ml min⁻¹. The resolution axis is split to indicate eluent concentrations where dramatic increases in separation occurred.

[Figure]

Fig S4. (a) Resolution of the six alkyl amine cation peak pairs using isocratic eluent methods at a flow rate of 0.75 ml min[-1].

(b) Resolution of the six alkyl amine cation peak pairs using isocratic eluent methods at a flow rate of 1.25 ml min[-1].

10

15

[Figure]

Fig S5. Calculated Van Deemter plots for the isocratic elutions of (a) $MMAH^+$ and (b) $TEAH^+$ at various MSA eluent concentrations and flow rates.

[Figure]

Fig S6. A chromatogram from an extracted $PM_{2.5}$ sample collected during a biomass-burning event in British Columbia at the Burnaby Kensington Park site.

Table S1. Comparison of methyl and ethyl amine external and standard addition calibration slopes and retention times ($t_r$)

| Analyte | External ($\mu S*min\ mol^{-1}$) | Standard addition ($\mu S*min\ mol^{-1}$) | Difference (%) | External $t_r$ range (min) | Standard addition $t_r$ range (min) |
|---|---|---|---|---|---|
| MMAH⁺ | 0.41E08 | 0.42E08 | 2 | 9.0 – 9.4 | 9.3 – 9.6 |
| DMAH⁺ | 0.98E08 | 1.01E08 | 3 | 10.4 – 10.8 | 10.6 – 11.0 |
| TMAH⁺ | 0.13E08 | 0.13E08 | 0 | 13.2 – 13.5 | 13.4 – 13.8 |
| MEAH⁺ | 0.47E08 | 0.51E08 | 8 | 10.1 – 10.5 | 10.3 – 10.8 |
| DEAH⁺ | 1.06E08 | 1.08E08 | 2 | 13.9 – 14.2 | 14.1 – 14.5 |
| TEAH⁺ | 0.57E08 | 0.57E08 | 0 | 23.6 – 24.0 | 23.9 -24.2 |

Table S2. Analytical performance of other IC methods used for the determination of atmospheric methyl and ethyl amines.

| Analyte | Detection method | Pre-conc | Column | LOD (pg) | Precision (%) | Reference |
|---|---|---|---|---|---|---|
| MMAH⁺ | CD | Yes | CS10 | 31 | 2 – 2.7 | Chang et al., 2003 |
| | CD | Yes | CS12A | 18 | 4.5 | VandenBoer et al., 2012 |
| | CD | No | CS14 | 2500 | 3.8 | Verriele et al., 2012 |
| | MS | No | CS14 | 500 | 5.8 | Verriele et al., 2012 |
| | CD | Yes | CS17 | 540 | 4.8 | VandenBoer et al., 2012 |
| | CD | Yes | CS19 | 30 - 650 | 5 | This work |
| | CD | No | Metrosep C2 | 21500 | 0.4 | Erupe et al., 2010 |
| | CD | Yes | Metrosep C4 | 2100 | 12.2 | Huang et al., 2014 |
| | CD | No | Metrosep C4 | 160 | 7.3 | Dawson et al., 2014 |
| DMAH⁺ | CD | Yes | CS10 | 40 | 2 – 2.7 | Chang et al., 2003 |
| | CD | Yes | CS12A | 25 | 1 | VandenBoer et al., 2012 |
| | CD | No | CS14 | 4000 | 10.5 | Verriele et al., 2012 |
| | MS | No | CS14 | 150 | 11.4 | Verriele et al., 2012 |
| | CD | Yes | CS17 | 870 | 14 | VandenBoer et al., 2012 |
| | CD | No | CS17 | 1500 | 1.2 | Li et al., 2009 |
| | CD | Yes | CS19 | 40 - 650 | 5 | This work |
| | CD | No | Metrosep C2 | 23000 | 1.4 | Erupe et al., 2010 |
| | CD | Yes | Metrosep C4 | 3800 | 15.7 | Huang et al., 2014 |
| | CD | No | Metrosep C4 | 320 | 1.1 | Dawson et al., 2014 |
| TMAH⁺ | CD | Yes | CS10 | 26 | 2 – 2.7 | Chang et al., 2003 |
| | CD | Yes | CS12A | 220 | 1 | VandenBoer et al., 2012 |
| | CD | No | CS14 | 2500 | N/A | Verriele et al., 2012 |
| | MS | No | CS14 | 500 | 12.2 | Verriele et al., 2012 |
| | CD | Yes | CS17 | 1580 | 3.3 | VandenBoer et al., 2012 |
| | CD | No | CS17 | 2000 | 3.5 | Li et al., 2009 |
| | CD | Yes | CS19 | 300 - 1200 | 16 | This work |

| | | | | | | |
|---|---|---|---|---|---|---|
| | CD | No | Metrosep C2 | 38000 | 1.1 | Erupe et al., 2010 |
| | CD | No | Metrosep C4 | 970 | 6.1 | Dawson et al., 2014 |
| MEAH$^+$ | CD | Yes | CS10 | 37 | 2 – 2.7 | Chang et al., 2003 |
| | CD | Yes | CS12A | 33 | 12 | VandenBoer et al., 2012 |
| | CD | No | CS14 | 1000 | 5.1 | Verriele et al., 2012 |
| | MS | No | CS14 | 500 | 7.9 | Verriele et al., 2012 |
| | CD | Yes | CS17 | 790 | 10 | VandenBoer et al., 2012 |
| | CD | Yes | CS19 | 200 - 700 | 7 | This work |
| | CD | Yes | Metrosep C4 | 2200 | 4.3 | Huang et al., 2014 |
| DEAH$^+$ | CD | Yes | CS12A | 195 | 14 | VandenBoer et al., 2012 |
| | CD | No | CS14 | N/A | N/A | Verriele et al., 2012 |
| | MS | No | CS14 | 35 | 9 | Verriele et al., 2012 |
| | CD | Yes | CS17 | 1140 | 3.5 | VandenBoer et al., 2012 |
| | CD | Yes | CS19 | 100 - 800 | 8 | This work |
| | CD | Yes | Metrosep C4 | 4100 | 4.6 | Huang et al., 2014 |
| TEAH$^+$ | CD | Yes | CS12A | 32000 | 2 | VandenBoer et al., 2012 |
| | CD | Yes | CS17 | 1870 | 5.9 | VandenBoer et al., 2012 |
| | CD | Yes | CS19 | 500 - 1400 | 12 | This work |
| | CD | Yes | Metrosep C4 | 15900 | 5.1 | Huang et al., 2014 |

Table S3. Mass loadings of amines and ammonium in size-resolved particle diameter ($D_p$) samples from an aged biomass burning plume sampled in St. John's, Newfoundland on July 6, 2013.

| $D_p$ | Mass loading (ng m$^{-3}$) | | | | | | |
|---|---|---|---|---|---|---|---|
| (µm) | NH$_4^+$ | MMAH$^+$ | DMAH$^+$ | TMAH$^+$ | MEAH$^+$ | DEAH$^+$ | TEAH$^+$ |
| 10 - 18 | BDL | $2.0 \pm 0.2$ | $0.6 \pm 0.2$ | BDL | BDL | $3 \pm 1$ | $2 \pm 1$ |
| 5.6 - 10 | BDL | BDL | $0.7 \pm 0.3$ | $2 \pm 2$ | BDL | $2.2 \pm 0.9$ | $2 \pm 1$ |
| 3.2 – 5.6 | BDL | $0.11 \pm 0.03$ | $0.4 \pm 0.1$ | BDL | BDL | BDL | BDL |
| 1.8 – 3.2 | BDL | $0.10 \pm 0.03$ | $0.25 \pm 0.09$ | $2 \pm 1$ | BDL | $1.4 \pm 0.6$ | $1.3 \pm 0.7$ |
| 1 – 1.8 | BDL | $2.9 \pm 0.8$ | $3 \pm 1$ | $3 \pm 2$ | BDL | $27 \pm 4$ | $4 \pm 2$ |
| 0.56 – 1 | $719 \pm 7$ | $1.4 \pm 0.4$ | $190 \pm 4$ | $5 \pm 3$ | $0.4 \pm 0.2$ | $1300 \pm 200$ | $2 \pm 1$ |
| 0.32 – 0.56 | $443 \pm 4$ | $11 \pm 3$ | $208 \pm 4$ | $4 \pm 3$ | $0.21 \pm 0.08$ | $1300 \pm 200$ | $4 \pm 2$ |
| 0.18 – 0.32 | $236 \pm 2$ | $6 \pm 2$ | $80 \pm 2$ | BDL | BDL | $560 \pm 90$ | $2 \pm 1$ |
| 0.10 – 0.18 | $30 \pm 50$ | $0.4 \pm 0.1$ | $30 \pm 10$ | BDL | $0.6 \pm 0.2$ | $200 \pm 30$ | BDL |

| 0.056 – 0.10 | 20 ± 30 | 3 ± 1 | 6 ± 2 | 4 ± 3 | BDL | 58 ± 9 | 3 ± 2 |
|---|---|---|---|---|---|---|---|
| 0.032 – 0.056 | 40 ± 70 | 0.11 ± 0.03 | 7 ± 3 | BDL | BDL | 49 ± 8 | BDL |
| 0.018 – 0.032 | BDL | 0.10 ± 0.03 | BDL | BDL | BDL | 4 ± 2 | BDL |
| 0.010 – 0.018 | BDL | 0.30 ± 0.08 | BDL | BDL | BDL | BDL | BDL |

BDL = below detection limits

Table S4. Mass loadings of amines and ammonium in a fresh biomass burning plume at Burnaby Kensington Park and North Vancouver/Second Narrows sites in British Columbia in July 2015. Relative time is calculated with respect to the local maximum $PM_{2.5}$ intrusion.

| Sampling site | Relative Time (h) | Mass loading (ng m$^{-3}$) | | | | |
|---|---|---|---|---|---|---|
| | | $NH_4^+$ | $iMPAH^+$ | $TMAH^+$ | $DEAH^+$ | $TEAH^+$ |
| BKP | - 24 | 70 ± 20 | 1.4 ± 0.9 | 0.6 ± 0.5 | 0.9 ± 0.5 | 0.2 ± 0.1 |
| BKP | - 16 | 90 ± 30 | 6 ± 4 | 3 ± 2 | 1.6 ± 0.8 | 0.08 ± 0.06 |
| BKP | - 8 | 130 ± 40 | BDL | 5 ± 4 | BDL | BDL |
| BKP | 0 | 90 ± 30 | 20 ± 10 | 9 ± 7 | 8 ± 4 | BDL |
| BKP | 8 | 90 ± 30 | 2 ± 1 | 2 ± 1 | 1.2 ± 0.7 | BDL |
| BKP | 16 | 80 ± 20 | BDL | BDL | BDL | BDL |
| BKP | 24 | 31 ± 9 | BDL | BDL | BDL | BDL |
| NVSN | - 24 | 130 ± 40 | 5 ± 3 | BDL | BDL | BDL |
| NVSN | - 16 | 400 ± 100 | 14 ± 9 | BDL | BDL | BDL |
| NVSN | 0 | 300 ± 100 | 40 ± 30 | BDL | BDL | BDL |
| NVSN | 8 | 300 ± 100 | 60 ± 40 | BDL | BDL | BDL |
| NVSN | 16 | 70 ± 20 | BDL | BDL | BDL | BDL |
| NVSN | 24 | 100 ± 30 | BDL | BDL | BDL | BDL |

BDL = below detection limits